# *N6*-methyladenosine (m6A) reader Pho92 is recruited co-transcriptionally and couples translation to mRNA decay to promote meiotic fitness in yeast

Radhika A Varier[1]*[†], Theodora Sideri[1†], Charlotte Capitanchik[1†], Zornitsa Manova[1], Enrica Calvani[1], Alice Rossi[1], Raghu R Edupuganti[2,3], Imke Ensinck[1], Vincent WC Chan[1], Harshil Patel[1], Joanna Kirkpatrick[1], Peter Faull[1,4], Ambrosius P Snijders[1], Michiel Vermeulen[2], Markus Ralser[1,5], Jernej Ule[1,6], Nicholas M Luscombe[1,7,8], Folkert J van Werven[1]*

[1]The Francis Crick Institute, London, United Kingdom; [2]Department of Molecular Biology, Faculty of Science, Radboud Institute for Molecular Life Sciences (RIMLS), Oncode Institute, Radboud University Nijmegen, Nijmegen, Netherlands; [3]Department of Human Genetics, University of Miami Miller School of Medicine, Sylvester Comprehensive Cancer Center, Biomedical Research Building, Miami, United States; [4]Biological Mass Spectrometry Facility, The University of Texas at Austin, Austin, United States; [5]Department of Biochemistry, Charité Universitätsmedizin Berlin, Berlin, Germany; [6]Dementia Research Institute, King's College London, London, United Kingdom; [7]Department of Genetics, Evolution and Environment, UCL Genetics Institute, London, United Kingdom; [8]Okinawa Institute of Science and Technology Graduate University, Okinawa, Japan

*For correspondence:
radhikaav@gmail.com (RAV);
folkert.vanwerven@crick.ac.uk
(FJvW)

[†]These authors contributed
equally to this work

Competing interest: The authors
declare that no competing
interests exist.

Reviewing Editor: Timothy W
Nilsen, Case Western Reserve
University, United States

**Abstract** *N6*- methyladenosine (m6A) RNA modification impacts mRNA fate primarily via reader proteins, which dictate processes in development, stress, and disease. Yet little is known about m6A function in *Saccharomyces cerevisiae*, which occurs solely during early meiosis. Here, we perform a multifaceted analysis of the m6A reader protein Pho92/Mrb1. Cross-linking immunoprecipitation analysis reveals that Pho92 associates with the 3'end of meiotic mRNAs in both an m6A-dependent and independent manner. Within cells, Pho92 transitions from the nucleus to the cytoplasm, and associates with translating ribosomes. In the nucleus Pho92 associates with target loci through its interaction with transcriptional elongator Paf1C. Functionally, we show that Pho92 promotes and links protein synthesis to mRNA decay. As such, the Pho92-mediated m6A-mRNA decay is contingent on active translation and the CCR4-NOT complex. We propose that the m6A reader Pho92 is loaded co-transcriptionally to facilitate protein synthesis and subsequent decay of m6A modified transcripts, and thereby promotes meiosis.

## Editor's evaluation

The authors identified and characterized an m6A reader protein Pho92 in *Saccharomyces cerevisiae*, providing several lines of evidence suggesting that it functions in RNA decay and translation, using a combination of molecular biological and computational approaches. The work is of interest to molecular biologists who are interested in understanding how mRNA modifications affect cell fate decisions.

## Introduction

Differentiation from one cell type to another requires accurate and timely control of gene expression. Molecular mechanisms of transcription, RNA processing and translation ensure precise temporal expression of genes. Over the last decade, it has become evident that the fate of cells is also largely determined through RNA modifications (*Roundtree et al., 2017*). Specifically, the *N6*-methyladenosine (m6A) modification on messenger RNAs (mRNAs) has been shown to play a pivotal role in cell differentiation, development, and disease pathology (*Yang et al., 2020*).

In mammals, the machinery that deposits m6A, also known as the m6A writer complex, consists of the catalytic subunit METTL3 and the catalytically inactive METTL14. Together they form the RNA binding groove, and are bound by WTAP and several other interacting proteins required to lay down the m6A mark on mRNAs (*Liu et al., 2014*). The m6A writer complex has strong preference for certain motifs (e.g. RRACH in mammals, and RGAC in yeast), and deposits m6A predominantly at the 3'end of transcripts (*Dominissini et al., 2012*; *Meyer et al., 2012*; *Schwartz et al., 2013*). Furthermore, the m6A mark is recognized and bound by reader proteins, which in turn recruit other protein complexes to control the fate of m6A marked transcripts (*Zaccara et al., 2019*). The YT521-B Homology (YTH) domain facilitates the m6A interaction of most m6A reader proteins, which defines the YTH family of proteins that is conserved from yeast to humans including plants (*Patil et al., 2018*). YTH domain containing proteins execute various mRNA fate functions, which includes mRNA decay, translation, transcription, and chromatin regulation (*Wang et al., 2014*; *Zhou et al., 2015*; *Xiao et al., 2016*; *Lasman et al., 2020*). Interestingly, YTH family proteins have been implicated both in translation and decay, suggesting that these proteins may exert overlapping roles in regulating gene expression (*Wang et al., 2014*; *Wang et al., 2015*; *Lasman et al., 2020*; *Zaccara and Jaffrey, 2020*).

In yeast, the m6A modification occurs during early meiosis, as part of a critical developmental program known as sporulation (*Shah and Clancy, 1992*; *Clancy et al., 2002*). During sporulation, diploid cells undergo a single round of DNA replication followed by two consecutive nuclear meiotic divisions to produce four haploid spores (*Figure 1A*). Specifically, the m6A modification occurs during early meiosis (meiotic entry, DNA replication, prophase), and declines once cells undergo meiotic divisions (*Schwartz et al., 2013*; *Figure 1A*). The deposition of m6A on mRNAs is catalysed by the methyltransferase Ime4, the Mettl3 orthologue. Ime4 also requires Mum2, the WTAP orthologue, and Slz1, which together comprise the m6A writer machinery in yeast called the MIS complex (*Agarwala et al., 2012*; *Figure 1A*). Depending on the strain background, cells lacking Ime4 are either completely or severely impaired in undergoing meiosis and sporulation (*Clancy et al., 2002*; *Hongay et al., 2006*). Some evidence suggests that m6A contributes to the decay and translation of mRNAs (*Bodi et al., 2015*; *Bushkin et al., 2019*). Yeast harbours one known m6A reader protein Pho92, also known as methylation RNA binding protein 1 (Mrb1), which has a conserved YTH domain that is required for its interaction with m6A (*Schwartz et al., 2013*; *Xu et al., 2015*). However, the molecular function of the m6A modification and m6A reader proteins in yeast remains largely unknown.

Here, we dissected the function of m6A-dependent RNA binding proteins. By employing proteomics and individual-nucleotide resolution UV cross-linking and immunoprecipitation (iCLIP) analysis we reveal that Pho92/Mrb1 binds at 3' ends of mRNAs in both an m6A dependent and independent manner. During early meiosis, Pho92 localizes to the nucleus, through its interaction with the RNA Polymerase II associated transcription elongation complex, Paf1C, and transitions to the cytoplasm to associate with actively translating ribosomes. The Pho92 deletion mutant displays decreased decay of m6A modified mRNAs as well as defects in protein synthesis. We propose that Pho92 is part of an mRNA fate control pathway for m6A modified mRNAs that is instated during transcription and couples protein synthesis to mRNA decay, and thereby promotes gamete fitness in yeast.

## Results

### Pho92/Mrb1 is a developmentally regulated m6A reader

To systematically identify proteins that interact with m6A in yeast, we incubated m6a modified RNA baits with cell extracts collected either pre-meiosis (PM) or during early meiosis (EM) (*Figure 1A and B*). We used the previously described synchronization method that relies on the induction of the master regulator Ime1 from the *CUP1* promoter (*pCUP-IME1*) to induce meiosis. We collected the PM sample at 2 hr in sporulation medium (SPO) and EM sample at 4 hr in SPO, when the MIS complex is

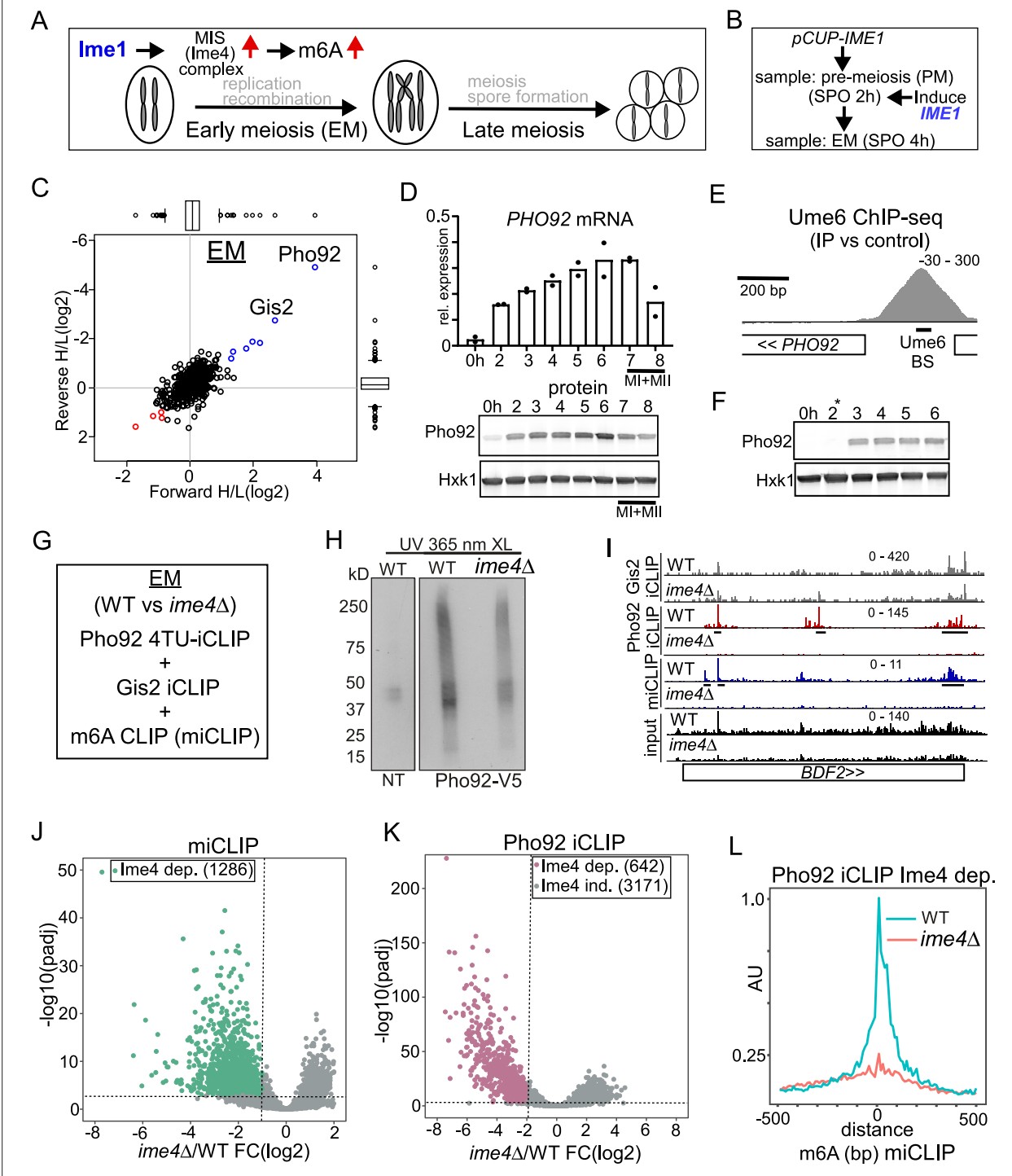

**Figure 1.** Pho92, but not Gis2, binds to m6A-modified transcripts. (**A**) Schematic overview of the yeast meiotic program. Ime1 induces the transcription of the MIS complex. m6A occurs during early meiosis. (**B**) Scheme describing set up for synchronized meiosis. Cells were grown in rich medium till saturation, shifted to SPO, and cells were induced to enter meiosis using *CUP1* promoter fused to *IME1* (*pCUP-IME1*, FW2444). Time points were taken at 2 hr and 4 hr. (**C**) Scatter plot displaying proteins identified in m6A consensus oligo pull down versus control. In short, cells were grown in rich medium till saturation, and shifted to SPO, and cells were induced to enter meiosis using *CUP1* promoter fused to *IME1* (*pCUP-IME1*, FW 2444). Protein extracts were incubated using m6A and control RNA oligo bound to streptavidin beads. Eluted proteins were differentially labelled with light and heavy dimethyl isotopes, mixed, and proteins from forward and reverse label swap reactions were identified by MS. (**D**) Pho92 expression prior and during meiosis. Diploid cells with Pho92 tagged with V5 (FW 4478) were induced to enter meiosis in sporulation medium (SPO). Samples were taken at the indicated time points, and Pho92 RNA and protein levels were determined by RT-qPCR and western blotting. n=2 biological repeats. (**E**) ChIP-seq data for Ume6

*Figure 1 continued on next page*

*Figure 1 continued*

at the *PHO92* locus. Indicated are the Ume6 binding site, and the ChIP-seq signal. Data were taken from **Chia et al., 2021**. (**F**) Pho92 expression prior to and after induction of Ime1 expression. Cells harbouring *pCUP-IME1* and Pho92 tagged with V5 (FW 9962) were induced to enter meiosis in sporulation medium (SPO). After 2 hr in SPO, Ime1 was induced with copper sulphate (labelled with *). Samples were taken at the indicated time points, and protein levels were determined by western blotting. Hxk1 was used as loading control respectively. n=2 biological repeats. (**G**) Experimental setup for Pho92 4TU-iCLIP, Gis2 iCLIP, and m6A CLIP (miCLIP) in wild-type (WT) and *ime4Δ* cells. At least n=3 biological repeats were performed. (**H**) Autoradiograph showing the protein RNA complexes in no tag (NT) and Pho92-V5 cells. In short, cells were grown till 4 hr in SPO in presence of 4-thiouracil. Cell were harvested and crosslinked. Protein extracts were generated, and Pho92 was immunoprecipitated with anti V5 antibodies. RNA-protein complexes were labelled with (γ-$^{32}$P)-ATP, and separated by SDS page, and transferred to nitrocellulose membrane. Displayed are the signals obtained for no tag control (FW4256), Pho92-V5 (WT, FW 4472), and Pho92-V5 in *ime4Δ* (FW4505). (**I**) Integrative genome browser (IGV) view of *BDF2* gene for Pho92, Gis2 iCLIP and miCLIP in WT and *ime4Δ* cells. Tracks are crosslink per million normalised, strand-specific bigWigs. (**J and K**) Volcano plots comparing WT and *ime4Δ* cells for miCLIP (**J**) and Pho92 4TU-iCLIP (**K**). The Ime4-dependent binding sites are labeled, as determined by a criteria of log2FoldChange –1 and adjusted p-value <0.001 for miCLIP and log2FoldChange <= –2 and adjusted p-value <0.001 for the iCLIP experiments. (**L**) Metagene analysis of Pho92 CLIP Ime4-dependent sites for WT and *ime4Δ* cells. Data was centred on the m6A sites identified with miCLIP.

The online version of this article includes the following source data and figure supplement(s) for figure 1:

**Source data 1.** Licor Odyssey multi-channel scan of western blot probed for Pho92-V5 and Hxk1 in *Figure 1D*.

**Source data 2.** Licor Odyssey multi-channel scan of western blot probed for Pho92-V5 and Hxk1 in *Figure 1F*.

**Source data 3.** Scan of Autoradiograph with RNA-protein crosslinks of Pho92-V5 in *Figure 1H*.

**Figure supplement 1.** Pho92, but not Gis2, binds to m6A marked transcripts in vitro and in vivo.

**Figure supplement 1—source data 1.** Scan of coomassie gel showing m6A pulldown with GST-Proteins in *Figure 1—figure supplement 1C*.

**Figure supplement 1—source data 2.** Scan of western blot - probed for Gis2-V5 showing pulldown of GST-Gis2 with control consensus in *Figure 1—figure supplement 1D*.

**Figure supplement 1—source data 3.** Scan of coomassie gel showing m6A pulldown with GST-Pho92 full length and truncations in *Figure 1—figure supplement 1E*.

**Figure supplement 1—source data 4.** Scan of Autoradiograph with RNA-protein crosslinks of Pho92-V5 in *Figure 1—figure supplement 1H*.

**Figure supplement 1—source data 5.** Scan of western blot showing IP of Pho92-V5 in *Figure 1—figure supplement 1H*.

**Figure supplement 2.** Pho92, but not Gis2, binds to m6A marked transcripts in vitro and in vivo.

---

induced by Ime1 and consequently m6A occurs (*Figure 1A and B*; *Agarwala et al., 2012*; *Schwartz et al., 2013*; *Chia and van Werven, 2016*).

We designed RNA baits consisting of four repeats of the canonical GGACU motif, that were either m6A modified (GGm6ACU) or unmodified (GGACU) (*Figure 1—figure supplement 1A*; *Edupuganti et al., 2017*). Subsequently, we performed differential labelling and mass-spectrometry (*Figure 1—figure supplement 1A*).

We identified two proteins, Pho92 and Gis2, showing enrichment of binding to m6a modified baits compared to the unmodified bait (log2 enrichment >2). The known m6A reader and YTH domain containing protein Pho92/Mrb1 was specifically enriched during EM, while Gis2 was enriched in both PM and EM lysates (*Figure 1C*, *Figure 1—figure supplement 1B*, *Supplementary file 4*; *Schwartz et al., 2013*; *Xu et al., 2015*). Gis2 is an RNA binding protein that associates with ribosomes to facilitate translation by interacting with translation initiation factors (*Rojas et al., 2012*). To validate whether Gis2 preferentially binds to m6A in vitro, we used recombinant proteins and performed RNA binding assays. Gis2 displayed no binding to either modified or unmodified oligos in a repeating DRACH motif sequence context, though Gis2 binding was detected with a control RNA oligo harbouring its known binding motif of GA(A/U) repeats (*Figure 1—figure supplement 1C and D*). However, consistent with other reports (*Schwartz et al., 2013*; *Xu et al., 2015*), Pho92 associated with the m6A modified RNA oligo, but not with the unmodified control RNA oligo (*Figure 1—figure supplement 1C*). As expected, Pho92 protein lacking the YTH domain (YTHΔ) completely abrogated binding to the m6A modified oligo, whilst removal of the amino-terminus (NΔ), which is largely unstructured, had no impact on m6A modified oligo binding (*Figure 1—figure supplement 1E*).

Given that Pho92 was enriched only in early meiosis in the RNA pulldown, we assessed the expression of Pho92 protein and mRNA in wild-type cells entering meiosis. In line with proteomic analysis, we found that Pho92 was expressed in early meiosis prior to when meiotic divisions took place (MI + MII) (*Figure 1D*). We examined whether the promoter of *PHO92* is possibly under direct control of Ume6, the DNA binding transcription factor that interacts with Ime1 to activate early meiosis gene

transcription (*Steber and Esposito, 1995*). We found that the *PHO92* promoter harbours a canonical Ume6 binding site (CGGCGGCTA) 230 nucleotides (nt) upstream of the start codon, which displays strong binding of Ume6 (*Chia et al., 2021*; *Figure 1E*). To assess whether indeed Pho92 was dependent on Ime1, we induced *IME1* from the *CUP1* promoter (*Figure 1B*). Prior to *pCUP1-IME1* induction Pho92 was not detectable, but Pho92 transcript and protein rapidly accumulated after *IME1* induction (*Figure 1F*, *Figure 1—figure supplement 1F*). Thus, Pho92 is expressed in early meiosis owing to its promoter being regulated by the meiosis-specific transcription factor Ime1 and its interacting partner Ume6.

## Pho92 associates with mRNAs in an m6A-dependent and independent manner

We next set out to determine the transcripts that Pho92 and Gis2 bind in vivo, their sequence and positional preferences, and to address whether this interaction is mediated by m6A. Specifically, we used a recently improved iCLIP protocol (*Lee et al., 2021*) to map Pho92 and Gis2 RNA binding sites during early meiosis (4 h SPO). Gis2 crosslinked well to RNA with UV irradiation at 254 nm, whereas Pho92 elicited better crosslinking to RNA with UV irradiation at 365 nm in the presence of uracil analog, 4-thiouracil (4TU-iCLIP, *Figure 1G, H*, *Figure 1—figure supplement 1H*; *Lee and Ule, 2018*). Yeast cells lacking the sole m6A methyltransferase Ime4 display no detectable m6A signal during meiosis (*Clancy et al., 2002*; *Schwartz et al., 2013*). Therefore, we performed iCLIP in both wild-type and *ime4Δ* cells, to distinguish between Ime4-dependent and independent binding. To further assess whether the Ime4-dependent binding equates to specific binding of the reader protein to m6A sites, we performed m6A individual-nucleotide-resolution cross-linking and immunoprecipitation (miCLIP) under the same growth conditions (*Linder et al., 2015*; *Lee et al., 2021*). By using our *ime4Δ* control, we were able to distinguish bona fide m6A peaks from non-specific antibody binding. To control for changes in RNA abundance between wild-type and *ime4Δ* conditions, we also produced matched input libraries from mRNAs used for the miCLIP experiments, which also matched the timepoint used for the iCLIP experiments (4 h SPO).

Due to the high proportion of cDNAs truncating at the peptide that is crosslinked to RNA fragments, we used the start positions of uniquely mapping reads to assign the positions of crosslink sites in iCLIP, and m6A sites in miCLIP (*Lee and Ule, 2018*). iCLIP and miCLIP replicate samples were highly reproducible at the level of counts per peak, clustering together in principle component analysis (PCA), with the greatest variance being due to the cells' genetic background (wild type (WT) or *ime4Δ*) (*Figure 1—figure supplement 1G*). However, it is notable that the proportion of variance accounted for by genetic background was much less for Gis2 (34%) than for the miCLIP (80%) or Pho92 (58%). For example, the *BDF2* transcript harboured several Pho92 binding sites that depended on Ime4, while Gis2 displayed m6A-independent binding (*Figure 1I*).

To identify m6A sites, we filtered for miCLIP peaks that were significantly reduced in *ime4Δ* compared to wild-type cells (log2FoldChange <= –1, adjusted p-value <0.001), and found 1286 m6A sites in 870 genes (*Figure 1J*). A larger number of Pho92 peaks (642 peaks in 507 genes, 16.7% of all detected peaks) were found to be reliably reduced in *ime4Δ* (log2FoldChange <= –2, adjusted p-value <0.001) (*Figure 1K*). Using a less stringent criteria, we could designate up to 30% (1130/3823 peaks at log2FoldChange ≤ 0, adjusted p-value <0.05) of detected Pho92 peaks as reduced in *ime4Δ*; however, for subsequent analysis, we will refer to the 642 stringently defined peaks as 'Ime4-dependent' Pho92 binding sites. This means that surprisingly, a large subset of Pho92 binding sites do not decrease in *ime4Δ* cells, indicating that Pho92 can also associate with transcripts in an Ime4-independent manner (*Figure 1K*). We ascertained that this is likely not background binding of Pho92 because in *ime4Δ*cells clear enrichment for Pho92-RNA complexes can be detected compared to the untagged control (*Figure 1H*, *Figure 1—figure supplement 1H*). Indeed, iCLIP of the untagged control had overall much less signal compared to the Pho92 iCLIP (*Figure 1—figure supplement 1I*).

We assessed whether Ime4-dependent Pho92 and Gis2 binding was also m6A-dependent. First, we calculated the distance to the nearest m6A site from each Ime4-dependent Pho92 binding site. To do so, we validated our miCLIP data against published datasets containing analyses of multiple time points during meiosis using different m6A-sequencing techniques (*Schwartz et al., 2013*; *Garcia-Campos et al., 2019*). Even though the time point of our miCLIP and the published datasets were not a match, m6A sites defined by miCLIP overlapped with m6A sites from the published datasets

with ~30% of miCLIP m6A sites being within 50nt of a published site (*Figure 1—figure supplement 2A, B*). We noticed that m6A modified transcripts detected by miCLIP showed an higher expression levels compared to m6A transcripts identified in the published dataset, which could explain the difference overlap between the two datasets (*Figure 1—figure supplement 2C*). The Ime4-dependent Pho92 sites mapped close to m6A sites as determined by miCLIP or published data (*Figure 1—figure supplement 2D, E*). Additionally, Ime4 dependent Pho92 association signal was highly enriched at the m6A sites we identified using miCLIP (*Figure 1L*). We conclude that a substantial portion of Pho92 binding is dependent on m6A.

In contrast, analysis of Gis2 iCLIP revealed 3563 peaks, of which only 43 peaks in 24 genes were decreased in *ime4Δ* representing only 1.2% of all Gis2 peaks (log2FoldChange <= –2, adjusted p-value <0.001) (*Figure 1—figure supplement 2F*). However, the few Ime4-dependent Gis2 binding events showed no clear proximity to m6A sites (*Figure 1—figure supplement 2G, H*). Taken together, our data demonstrate that Pho92 is an m6A reader protein in early yeast meiosis. For the remainder of this study we focussed on revealing the mechanism by which Pho92 controls the fate of m6A marked transcripts during yeast meiosis.

## Features of transcripts bound by Pho92

Upon closer inspection of the Pho92 iCLIP data, we found that Pho92 iCLIP peaks are found in the mRNAs of key regulators of early meiosis such as *IME1* and *RIM4* (*Figure 2A*). The *IME1* and *RIM4* Pho92 iCLIP peaks overlapped with m6A sites identified with miCLIP (*RIM4*), or, in the case of *IME1*, with comparative Nanopore sequencing of WT versus *ime4Δ* described previously (*Leger et al., 2021*; *Figure 2—figure supplement 1A*). As an exception, *IME2*, a key kinase in meiosis, displayed no detectable Pho92 binding, but, consistent with previous reports, it did however display an Ime4-dependent peak in miCLIP data, albeit below our stringent thresholds (log2FC = –0.6, padj = 0.007) (*Bodi et al., 2010*; *Schwartz et al., 2013*; *Figure 2B*). There was a good overlap (more than 50%) between Pho92 binding sites and m6A sites (*Figure 2C*). As noted for *IME2*, there were also m6A sites where Pho92 was not detected, which is likely technical in nature as our current data coverage is not sufficient to cover all m6A sites with either miCLIP or Pho92 iCLIP, and additional variation could be caused due to technical differences between miCLIP and iCLIP protocols.

Using STREME, we analysed the RNA sequence context of Ime4-dependent Pho92 binding. Strikingly, the highest ranked motif found in Ime4-dependent Pho92 binding sites was nearly identical to the highest ranked motif found for the m6A sites, which highlights again that the Ime4-dependent sites are mostly m6A-dependent sites (*Figure 2D*, *Figure 2—figure supplement 1B, C*). This motif is an 'extended' RGAC sequence context, containing upstream AN and downstream NNU nucleotides that were previously described in *Schwartz et al., 2013*. The Ime4-independent binding sites of Pho92 showed no enrichment for the m6A motif sequence but showed weak enrichment for short motif sequences containing GU dinucleotides, indicating that m6A-independent binding of Pho92 is conceivably driven independent of sequence context (*Figure 1H*, *Figure 2—figure supplement 1B, C*).

As previously stated, consistent with the m6A pattern at mRNAs, Pho92 binding was detected predominantly at the 3' end of transcripts, with 23% (145/642) of Ime4-dependent binding sites and 13% (410/3171) of Ime4-independent binding sites directly overlapping a STOP codon (*Figure 2E*, *Figure 2—figure supplement 1D*). We also found Pho92 binding sites in the protein coding sequence or at the 5' end of transcripts, which included key meiotic mRNAs (*IME1* and *RIM4*) (*Figure 2A*, *Figure 2—figure supplement 1D*). Although most transcripts contained just one m6A and/or a Pho92 binding site, some transcripts did contain two or more m6A and/or Pho92 binding sites (*Figure 2F and G*). The 3' end binding of Pho92 is at least partially independent of m6A and the m6A consensus sequence motif (*Figure 2E, H*). The Ime4-independent binding sites also showed positional enrichment at the 3' end of mRNA transcripts, suggesting that these are likely bona fide binding sites (*Figure 2E*, *Figure 2—figure supplement 1D*). Thus, Pho92 associates with transcripts predominantly at their 3' ends, and a subset of this binding occurs in an Ime4-dependent manner at m6A sites within canonical m6A motifs.

We performed gene ontology (GO) analysis of Pho92-bound transcripts (Ime4- dependent) and found that MAPK signalling, cell cycle and meiosis were the top enriched terms (*Figure 2I*). Notably, examination of m6A sites showed enrichment for MAPK signalling and cell cycle but not meiosis

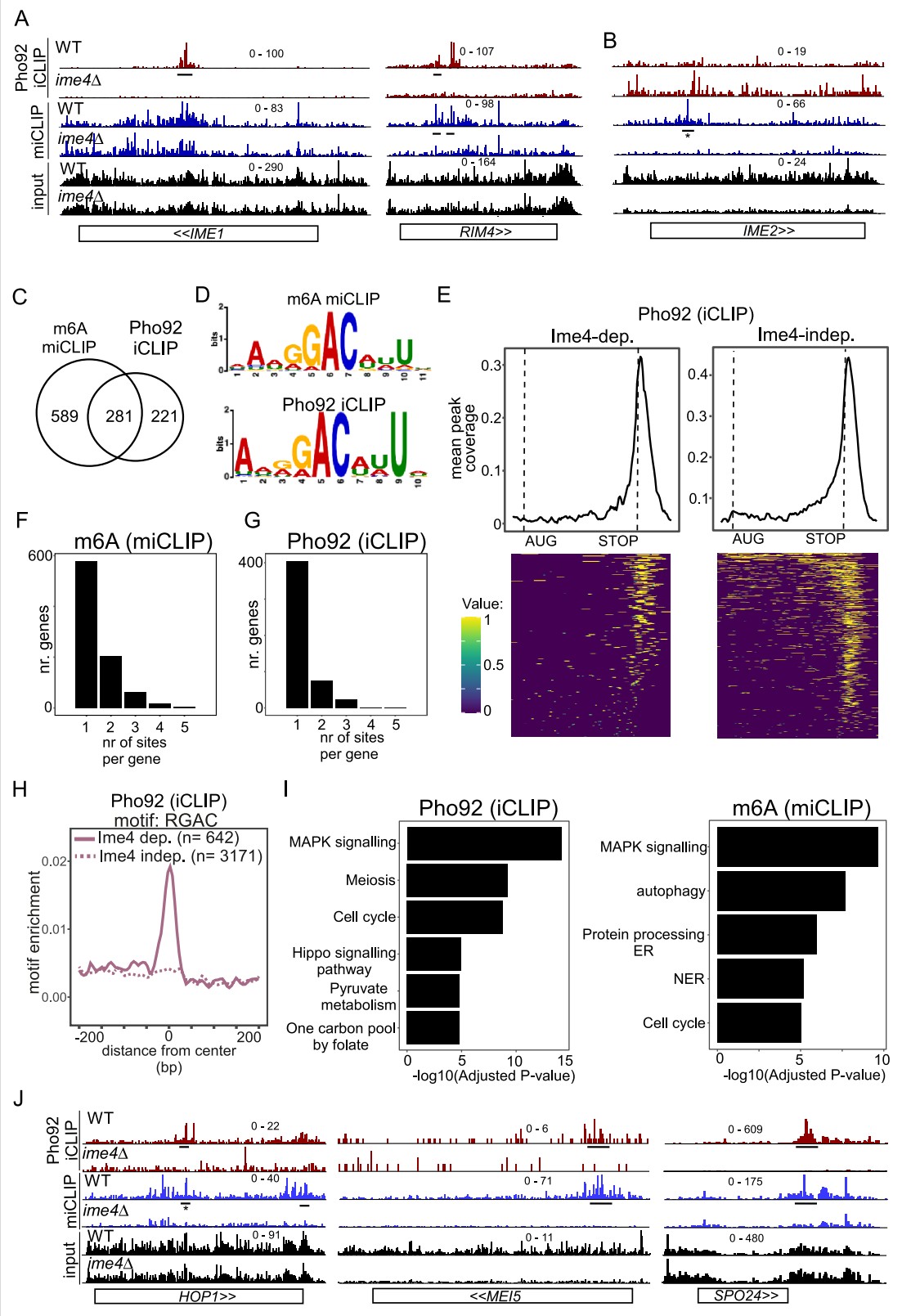

**Figure 2.** Features of transcripts bound by Pho92. (**A**) *IME1* and *RIM4* transcripts bound by Ime4-dependent Pho92. Shown are the crosslinks per million normalised bigWigs for Pho92 iCLIP and miCLIP in WT and *ime4Δ* cells. Underlined are the binding sites identified in the analysis. (**B**) Similar as A, except that *IME2* locus is displayed, which exhibits an m6A peak but no Pho92 binding. The m6A site is labelled. * this was reported in our analysis below the significance threshold (log2FC = −0.6, padj = 0.007). (**C**) Venn diagram showing the overlap between genes with m6A sites and Ime4-dependent Pho92

*Figure 2 continued on next page*

*Figure 2 continued*

binding. (**D**) Sequence logos of the top ranked motifs from m6A sites and Ime4-dependent Pho92 sites as determined by STREME. (**E**) Pho92 metagene profile split into genes containing Ime4-dependent Pho92 binding sites (left) and Ime4-independent sites (right). The matrix underlying the heatmap is scored 1 for binding site and 0 for no binding, intermediate values between 0 and 1 are due to smoothing in the visualisation. (**F**) Number of m6A sites (miCLIP) per transcript. On the x-axis transcripts with 1, 2, 3, 4, or 5 bindings sites. On the y-axis the number of genes for each category is displayed. (**G**) Similar to F, except that Pho92 binding sites were analysed. (**H**) Motif enrichment around m6A sites and Pho92 Ime4-dependent and independent binding sites for RGAC. Motif frequency plotted around the centre of Pho92 binding sites, split into Ime4-dependent sites (solid line) and Ime4-independent sites (dotted line). (**I**) Gene ontology (GO) enrichment analysis for Pho92-bound transcripts (left) and m6A harbouring transcripts based on the miCLIP analysis (right). On the y-axis the category of processes involved, while on x-axis the -log10(adjusted p-value) is displayed. (**J**) *HOP1, MEI5, and SPO24* transcripts bound by Pho92 and marked with m6A. Data tracks are crosslinks per million normalised bigWigs, visualised in IGV. Underlined are the binding sites identified in the analysis. * this was reported in our analysis below the significance threshold (log2FC = −0.94, padj = 0.004).

The online version of this article includes the following figure supplement(s) for figure 2:

**Figure supplement 1.** Features of transcripts bound by Pho92.

---

(*Figure 2I*), suggesting that m6A by itself regulates a different subset of transcripts than Pho92. Genes that showed Ime4-dependent Pho92 binding and corresponding m6A sites included those involved in regulating the meiotic program (such as *IME1*, *RIM4* and *SPO24),* DNA recombination and double strand-break repair (such as *HOP1* and *MEI5*) (*Figure 2J and A*, *Figure 2—figure supplement 1A*).

## Pho92 is important for the onset of meiosis and fitness of gametes

The yeast meiotic program is regulated by the master regulatory transcription factors, Ime1 and Ndt80 (*Kassir et al., 1988*; *Chu et al., 1998*). Ime1 drives early meiosis, while Ndt80 regulates the genes important for late meiosis, which includes meiotic divisions and spore formation (*Figure 1A*; *van Werven and Amon, 2011*). GO-analysis of Pho92 bound transcripts revealed that Pho92 associates with genes important for meiosis. This posed the question whether Pho92 acts at the same genes as Ime1 and Ndt80, or perhaps regulates a different category of genes, independent of these two transcription factors. To identify Ime1 and Ndt80 regulated genes we performed RNA-seq in *ime1Δ* and *ndt80Δ* cells in early meiosis (4 hours in SPO). As expected, the genes downregulated in *ndt80Δ* showed strong enrichment for the Ndt80 motif sequence, indicating the RNA-seq was able to identify direct target genes (*Figure 3—figure supplement 1A and B*; *de Boer and Hughes, 2012*). The promoter of transcripts downregulated in *ime1Δ*, however, showed enrichment for both Ndt80 and Ume6 motifs, likely because *ime1Δ* blocks induction of genes indirectly regulated by Ime1 (*Figure 3—figure supplement 1A*). We compared the genes that were significantly downregulated in *ime1Δ* and *ndt80Δ*cells to Pho92-bound transcripts. While the promoters of Pho92 bound transcripts (iCLIP) were enriched for the Ume6 motif, we found that only 81 genes out of the 507 Pho92 targets overlapped with the 1133 Ime1 regulated transcripts (*Figure 3A*, *Figure 3—figure supplement 1A*). Moreover, 54 genes overlapped with the 695 Ndt80 regulated transcripts (*Figure 3A*). Thus, a large fraction of transcripts bound by Pho92 were likely not targets of Ime1 or Ndt80. Similarly, little overlap was observed between m6A-marked transcripts and transcripts regulated by Ime1 or Ndt80 (*Figure 3—figure supplement 1C*). Our data suggest that m6A and Pho92 regulate a distinct subset of transcripts or genes than the transcription factors (Ime1 and Ndt80) that control the meiotic program.

Given that Pho92 is specifically expressed during early meiosis (*Figure 1D*), we expect that, like Ime4, Pho92 plays a role in controlling gametogenesis. Therefore, we closely examined the effects of *pho92Δ*on meiosis and spore formation in the SK1 strain background, widely used for studying meiosis and sporulation. We found that *pho92Δ*exhibits a delay in the onset of meiosis compared to WT, although much milder than *ime4Δ*(*Figure 3B*). The delay in meiotic divisions observed in *pho92Δ* cells was not alleviated by inducing *pCUP-IME1* (*Figure 3C*), supporting the idea that Pho92 and Ime1 regulate different subsets of genes. We also assessed whether Ndt80 expression can alleviate the effect of *pho92Δ* on meiosis by expressing Ime1 and Ndt80 together when cells were starved in sporulation medium (SPO) (*Figure 3D*). We induced Ndt80 from the *GAL1-10* promoter (*pGAL-NTD80*) using the transcription factor Gal4 fused to estrogen receptor (Gal4ER) and addition of β-estradiol (*Benjamin et al., 2003*; *Chia and van Werven, 2016*). Expressing Ime1 and Ndt80 together surprisingly had little effect on meiosis, as most cells (more than 80%) completed at least one meiotic division by 8 hr post-induction (WT, *Figure 3D*). Approximately 50% of *pho92Δ*cells completed one meiotic

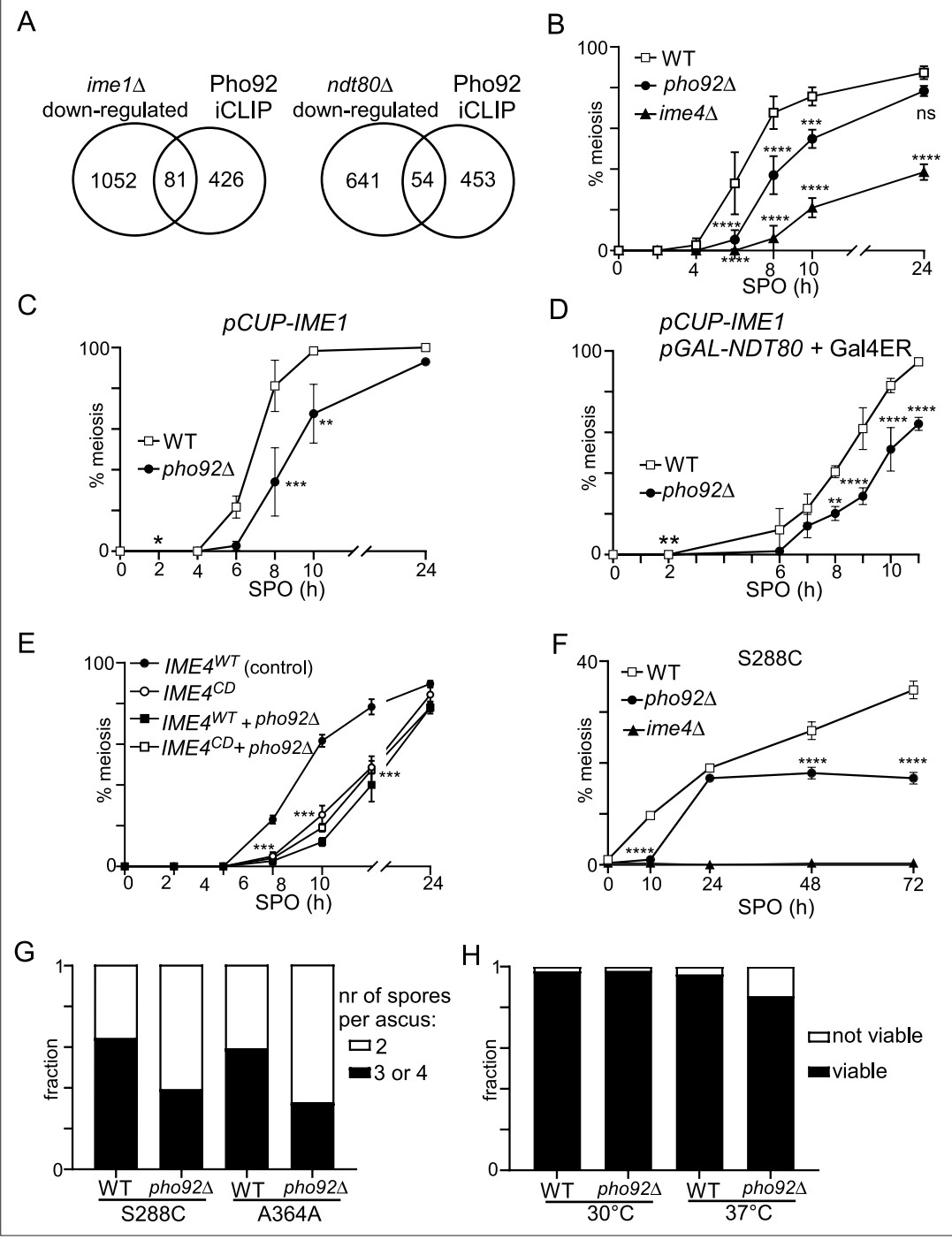

**Figure 3.** Pho92 is important for meiosis and fitness of gametes. (**A**) Venn diagram displaying the comparison between RNA-seq of *ime1Δ* (FW81) and *ndt80Δ* (FW 4911), and transcripts bound by Pho92. For this analysis, Ime1 and Ndt80-dependent genes were selected by taking the transcripts that were significantly down-regulated compared to the WT control. n=3 biological repeats. (**B**) Onset of meiosis in WT, *pho92Δ*, and *ime4Δ* cells (FW1511, FW3528, and FW725). Cells were grown in rich medium till saturation, shifted to pre-sporulation medium and grown for an additional 16 hr. Subsequently, cells were shifted to SPO, and samples were taken at the indicated time points. Cells were fixed, and stained, DAPI masses were counted for at least n=200 cells per biological repeat (n=3). Cells with two or more DAPI masses were considered as undergoing meiosis. (**C**) Similar as B, except that cells were induced to undergo meiosis using *pCUP-IME1* (WT FW2444, *pho92Δ* FW3576). Cells were grown as described in B, shifted to SPO, and after 2 hr treated with copper sulphate to induce *IME1* expression. (**D**) Similar as C, except that these also harboured *NDT80* under control of the *GAL1-10* promoter

*Figure 3 continued on next page*

*Figure 3 continued*

and fused Gal4 activation domain plus estrogen receptor (Gal4ER). Ndt80 expression was induced at 2 hr in SPO with β-estradiol together with Ime1 expression (WT FW2795, *pho92Δ* FW9070). (**E**) Similar analysis as B, except that strains harbouring *IME4^{WT}*(FW8736), *IME4^{WT} + pho92Δ*(FW11001), *IME4^{CD}* (FW8773), or *IME4^{CD} + pho92Δ* (FW10998) were used for analysis. The error bars represent the standard error of the mean (SEM) of n=3 biological repeats. Indicated is the highest *p*-value of the following comparisons: *IME4^{WT} + pho92Δ*or *IME4^{CD}* or *IME4^{CD} + pho92Δ* versus *IME4^{WT}* (***p<0.001) on a two-way ANOVA followed by a Fisher's least significant difference (LSD) test. (**F**) Similar analysis as B, except that A364A was used for the analysis (WT FW1671, *pho92Δ* FW8912, *ime4Δ* FW8913). The error bars represent SEM of n=3; *p<0.05, **p<0.01, ***p<0.001, ****p<0.0001, compared to WT control on a two-way ANOVA followed by a Fisher's least significant difference (LSD) test. n=3 biological repeats. (**G**) Spore packaging was assessed in WT and *pho92Δ* strains (S288C and A364A background). Number of packaged spores per ascus were counted for at least 200 asci. (**H**) Spore viability of WT (FW 1511) and *pho92Δ* (FW3531). Cells were patched from YPD agar plates to SPO agar plates and incubated for 3 days at 30 °C or 37 °C. Subsequently tetrads were dissected, and spores grown on YPD agar plates. The fraction of viable and not viable spores is indicated. At least n=150 spores for each condition (30 °C or 37 °C) and each strain (WT and *pho92Δ*) were used for the analysis.

The online version of this article includes the following figure supplement(s) for figure 3:

**Figure supplement 1.** Pho92 is important for meiosis and fitness of gametes.

**Figure supplement 2.** Pho92 is important for meiosis and fitness of gametes.

division after 8 hr of Ime1/Ndt80 induction (**Figure 3D**). Thus, the delay in meiosis in *pho92Δ*cells was not dependent on the expression of Ime1 and Ndt80.

One plausible explanation for the delay in meiotic divisions in *pho92Δ*cells is that Pho92 contributes to respiratory growth. Cells compromised in respiration enter meiosis inefficiently or not at all (**Jambhekar and Amon, 2008**; **Weidberg et al., 2016**). However, we found that *pho92Δ*cells showed no growth defect and had comparable doubling times (approximately 140 min) when acetate was used as the sole carbon source (**Figure 3—figure supplement 2A**).

In yeast, the catalytic inactive mutant of Ime4 (*IME4^{CD}*) has a milder phenotype than the *ime4Δ*, which perhaps explains the extent of the delay observed in *pho92Δ* (**Figure 3E**; **Agarwala et al., 2012**). Indeed, we found that *pho92Δ* and *IME4^{CD}* single and double mutants showed a similar delay in the onset of meiosis, suggesting the Pho92 and Ime4 catalytic functions phenocopy. We also tested whether Pho92 phenotype in meiosis is mediated by the YTH domain by generating a point mutation in the YTH domain, which we expressed from the *CUP1* promoter (*pCUP-PHO92^{W177A}*). We found that the onset of meiosis was delayed in *pCUP-PHO92^{W177A}* compared to control, indicating that Pho92 binding to m6A promotes fitness of meiosis (**Figure 3—figure supplement 2B**).

We also assessed whether Pho92 contributes to the onset of meiosis in strain backgrounds that sporulate with a lower efficacy (S288C and A364A). Both S288C and A364A displayed more than 50 percent less meiotic cells after 72 hours in SPO (**Figure 3F**, **Figure 3—figure supplement 2C**). We further found that Pho92 contributes to the packaging of spores because the fraction of tetrads per asci decreased and the fraction of dyads increased in *pho92Δ*cells (**Figure 3G**). We observed a modest reduction of spore viability when cells underwent sporulation at elevated temperatures (37 °C) (**Figure 3H**). We conclude that the m6A reader Pho92 is important for the fitness of meiotic progeny in yeast.

## Paf1C retains Pho92 in the nucleus during early meiosis

YTH domain containing proteins are conserved from humans to plants, and exercise various functions to control the fate of m6A transcripts (**Patil et al., 2018**). Little is known about the role of Pho92 in regulating the fate of m6A modified transcripts. To identify a possible function for Pho92 in yeast, we determined its protein-protein interactions. We performed immunoprecipitation of Pho92 followed by label-free quantitative mass spectrometry. In addition to purifying Pho92 from cells staged in early meiosis, we also expressed Pho92 from heterologous promoters to intermediate levels (*CYC1* promoter, *pCYC1*-FLAG-Pho92) in cycling cells (**Figure 4A**, **Figure 4—figure supplement 1A and B**). The purification of Pho92 from cells staged in early meiosis identified few interacting proteins and weak enrichment for Pho92 itself, likely because of the high protein turnover in extracts of meiotic cells (**Figure 4—figure supplement 1A and C**). In contrast, the purification from cycling cells grown in YPD

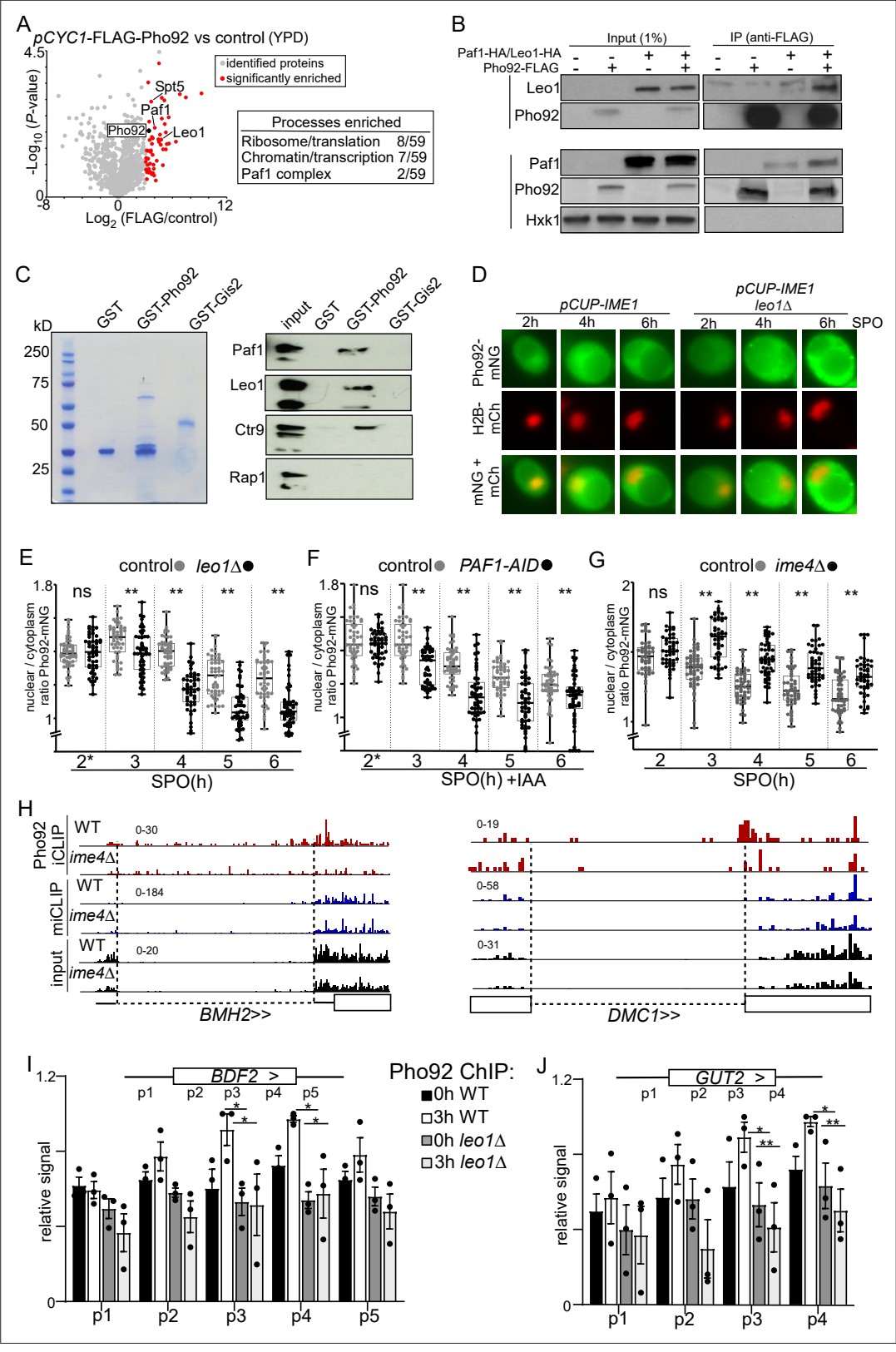

**Figure 4.** Paf1C interacts with Pho92 to direct Pho92 to nucleus. (**A**) Volcano plot IP-MS of Pho92 (left). Cells expressing the *CYC1* promoter (*pCYC1*-FLAG-Pho92, FW8734) controlling Pho92 expression and a FLAG-tag at the amino terminus were grown in rich medium conditions. Cell extracts from *pCYC1*-FLAG-Pho92 and control (FW629) were incubated with anti-FLAG beads and eluted with FLAG peptides and analysed by MS using label

*Figure 4 continued on next page*

*Figure 4 continued*

free quantification method. Significantly enriched proteins are displayed in red. Table (right) proteins enriched in IP-MS experiment (also data from plots in *Figure 4—figure supplement 1*). Highlighted are the processes and protein complexes enriched. (**B**) Co-immunoprecipitation of Pho92 and Paf1C. We used strains where *pPYK1*-FLAG-Pho92 was expressed with or without tagged HA-Paf1 or HA-Leo1 (FW9880 and FW9791). Cells harbouring FLAG-Pho92, HA-Paf1, or HA-Leo1 only were used as control (FW8732, FW9782, and FW9763). As a negative, control membranes were also probed for Hxk1, which does not interact with Pho92. (**C**) Pull-down of Paf1C by GST-Pho92, GST-Gis2 or GST alone induced in bacteria. Cell extracts were prepared and immobilized on Glutathione-agarose beads. GST-Pho92 bound to beads were subsequently incubated with extracts expressing HA-tagged Paf1, Leo1, Crt9, and Rap1 (FW9782, FW9763, FW9784, FW 4948). (**D**) Localization of Pho92 during entry in meiosis control and *leo1Δ* cells (FW9633 and FW9736). Cells expressing Pho92 fused with mNeongreen (Pho92-mNG) were used for the analysis. To determine nuclear localization, we used histone H2B fused to mCherry (H2B-mCh) and the cells also harboured *pCUP-IME1* to enable induction of synchronous meiosis. Cells were treated with copper sulphate at 2 hr (labelled with *), and samples were taken at the indicated time points. (**E**) Quantification of nuclear over cytoplasmic signal for Pho92-mNG in WT and *leo1Δ* (FW9633 and FW9736). Cells were grown as described in D. At least 150 cells were quantified per time point. Each datapoint is shown in addition to the box plots. ** $p<0.01$ Welch's paired t test. (**F**) Similar to E, except with Paf1 depletion strain. Paf1 is fused to an auxin induced degron (AID-tag), which is induced by copper sulphate and IAA treatment at 2 hr (FW10128). These cells also express *TIR1* ligase under control of the *CUP1* promoter. Cells harbouring the *TIR1* ligase alone were used as controls (FW10129). (**G**) Same as in E, except that *ime4Δ* cells were used for the analysis (FW9604). (**H**) Pho92 binding overlaps with *BMH2* and *DMC1* introns. Data tracks of Pho92 iCLIP, miCLIP, and miCLIP-input in WT and *ime4Δ* cells. Intron regions are designated with dashed lines. (**I and J**) ChIP-qPCR of Pho92 at the *BDF2* and *GUT2* loci during entry into meiosis in WT (FW4478) and *leo1Δ* (FW10113) cells. Biological repeats of WT and *leo1Δ* cells were grown in parallel, each sample was input normalized, subsequently the primer pair with highest signal was set to 1 for each biological repeat, which was primer pair p4 for 3 h WT *BDF2* and *GUT2*. The relative mean signal of n=3 biological repeats are displayed. *$p<0.05$ and ***$p<0.01$, on a two-way ANOVA followed by a Fisher's least significant difference (LSD) test.

The online version of this article includes the following source data and figure supplement(s) for figure 4:

**Source data 1.** Scan of western blot - probed for Paf1 (HA) in *Figure 4B*.

**Source data 2.** Scan of western blot - probed for Flag-Pho92 (reprobe of Paf1 (HA) blot) in *Figure 4B*.

**Source data 3.** Scan of western blot - probed for Paf1 (HA) in *Figure 4B*.

**Source data 4.** Scan of western blot - probed for Flag-Pho92 (reprobe of Paf1 (HA) blot) in *Figure 4B*.

**Source data 5.** Scan of western blot -probed for Hxk1 (control)- (reprobe of Paf1 (HA) blot) in *Figure 4B*.

**Source data 6.** Scan of coomassie gel showing expression of GST proteins in *Figure 4C*.

**Source data 7.** Scan of western blot with high exposure ECL detection probed for Paf1 (HA) and Leo1 (HA) in *Figure 4C*.

**Source data 8.** Scan of western blot with low exposure ECL detection probed for Ctr9 (HA) in *Figure 4C*.

**Source data 9.** Scan of western blot with medium exposure ECL detection probed for Control Rap1(HA) in *Figure 4C*.

**Figure supplement 1.** Paf1C interacts with Pho92 to direct Pho92 to nucleus.

**Figure supplement 1—source data 1.** Scan of western blot showing IP of Pho92-V5 in *Figure 4—figure supplement 1A*.

**Figure supplement 1—source data 2.** Scan of western blot showing IP of Flag-Pho92 in *Figure 4—figure supplement 1B*.

**Figure supplement 2.** Paf1C interacts with Pho92 to direct Pho92 to nucleus.

**Figure supplement 2—source data 1.** Licor Odyssey multi-channel scan of western blot probed for Paf1-AID and Hxk1 in *Figure 4—figure supplement 2G*.

**Figure supplement 3.** Paf1C interacts with Pho92 to direct Pho92 to nucleus.

**Figure supplement 3—source data 1.** Licor Odyssey multi-channel scan of western blot probed for Pho92-V5 and Hxk1 in *Figure 4—figure supplement 3C*.

---

identified large set of proteins that co-purified with Pho92 (*Figure 4—figure supplement 1B*). Specifically, several proteins involved in ribosomal biogenesis and translation were enriched compared to the untagged control (*Figure 4A* and *Supplementary file 6*). Surprisingly, we also found that Pho92 interacted with proteins involved in chromatin organization and transcription. These included two

subunits of RNA Polymerase II (pol II)-associated transcription elongation complex Paf1C (Leo1 and Paf1), as well as the highly conserved and essential pol II-associated transcription elongation factor Spt5 (*Figure 4A*).

To validate the interaction between Pho92 and Paf1C, we co-expressed epitope-tagged Pho92 and Paf1C and performed co-immunoprecipitation of Pho92 with Paf1C subunits Leo1 and Paf1 followed by western blotting. We found that Leo1, and to a lesser extent Paf1, co-immunoprecipitated with Pho92, but not the negative control hexokinase 1 protein (*Figure 4B*). Additionally, we affinity purified Pho92 fused to GST from *E. coli* and determined whether yeast extracts with HA-tagged Paf1, Leo1 or Ctr9 associated with recombinant GST-Pho92. We found that Paf1C subunits (Paf1, Leo1, and Ctr9) were enriched in the GST-Pho92 pull-down compared to the GST only or GST-Gis2 negative control, while the transcription factor Rap1 did not show enrichment (*Figure 4C*). We conclude that Pho92 interacts with Paf1C in vitro and in vivo.

Paf1C associates with pol II to promote transcription elongation and chromatin organization during transcription (*Rondón et al., 2004*; *Van Oss et al., 2017*). If Pho92 and Paf1C interact during meiosis, we hypothesized that Pho92 must localize to the nucleus and possibly associate with gene bodies during transcription. To address this, we first determined whether the Paf1C regulates Pho92 localization in cells entering meiosis. To visualize Pho92 we tagged the protein with mNeongreen (Pho92-mNG) and monitored Pho92-mNG localization during a synchronized meiosis (*pCUP-IME1*) and in wild type cells. We determined nuclear Pho92 signal using histone H2B fused to mCherry (H2B-mCh) (*Figure 4D* and *Figure 4—figure supplement 2A–C*). A large fraction of Pho92-mNG signal was in the nucleus prior to induction of meiosis (0 hr and 2 hr) and in the early time points of meiosis (3 and 4 hr) (*Figure 4D and E* (see WT)). At the later time points (5 and 6 hr), Pho92-mNG was however, largely excluded from the nucleus (*Figure 4D and E* (see control)). It is noteworthy that the Pho92-mNG whole cell signal was lower prior to induction of meiosis (*Figure 4—figure supplement 2D–2F*). A similar Pho92 localization and expression pattern was observed in cells WT for *IME1* (*Figure 4—figure supplement 2A–2B*). We conclude that Pho92 localization is dynamically regulated in cells undergoing early meiosis.

Next, we examined whether Paf1C contributed to Pho92 localization to the nucleus. We visualized Pho92-mNG localization in *leo1Δ* cells and in Paf1 depleted cells using the auxin induced degron (*PAF1-AID*) (*Figure 4—figure supplement 1G*). From 3 hr into sporulation onwards, we found that Pho92 was less nuclear in *leo1Δ* cells compared to the control (*Figure 4D and E*). A comparable difference in nuclear localization of Pho92 was observed when we depleted Paf1 (*PAF1-AID*+IAA) (*Figure 4F* and *Figure 4—figure supplement 2G*). We also examined whether Pho92 localization was dependent on the m6A writer Ime4. In *ime4Δ* cells Pho92 was retained more in the nucleus, indicating that m6A modified transcripts normally facilitate its exit from the nucleus (*Figure 4G*). We conclude that Pho92 localization is regulated throughout early meiosis. While Paf1C contributes to localization of Pho92 to the nucleus, m6A-modified transcripts facilitate transition of their reader Pho92 into the cytoplasm.

## Paf1C directs Pho92 to target genes

Interaction between Leo1 and nascent mRNA has been reported to stabilize the association of Paf1C with transcribed genes (*Dermody and Buratowski, 2010*). Our data showing the effect of Paf1C on Pho92 localization to the nucleus raised the question of whether Pho92 is recruited to pre-mRNAs during transcription via Paf1C. To corroborate this idea that Pho92 associates with nascent mRNAs, we scanned the iCLIP data for Pho92 association with intron regions which are typically retained in nascent mRNAs. We found 55 transcripts with Pho92 iCLIP signal in the intronic regions. For example, *BMH2* and *DMC1* intron regions contained Pho92 iCLIP crosslinks, while miCLIP and input displayed no detectable crosslinking within the intron sequence (*Figure 4H*, *Figure 4—figure supplement 3A*).

To substantiate the observation that Pho92 associates with some of its targets during transcription, we performed Pho92 chromatin immunoprecipitation (ChIP) qPCR across two genes that contained Ime4-dependent Pho92 binding sites (*BDF2* and *GUT2*) (*Figure 1I* and Figure 6I). In wild-type cells prior to entering meiosis (0 hr SPO), we found some Pho92 enrichment at these two ORFs (*Figure 4I and J*). The observed enrichment is likely not mediated via RNA because ChIP samples were treated with RNase. Cells entering meiosis (3 hr SPO) showed an increase in Pho92 binding. Importantly, in *leo1Δ* cells Pho92 ChIP signal over the whole gene body was reduced (*Figure 4I and J*). This decrease

cannot be attributed to expression differences because *leo1Δ*cells expressed *BFD2*, *GUT2*, and Pho92 to comparable levels as wild-type cells (*Figure 4—figure supplement 3B and C*). Likewise, *leo1Δ*cells entered meiosis with a mild delay (*Figure 4—figure supplement 3D*). Also, the Paf1C deletion or depletion mutants we used for our experiments had little effect on cellular m6A levels (*Figure 4—figure supplement 3E*). Thus, Pho92 associates with target genes in a Paf1C-dependent manner. Our data are consistent with a model where Pho92 is loaded co-transcriptionally to nascently produced mRNAs. However, it is also possible that Pho92 has a regulatory function in the nucleus involving chromatin and transcription as has been reported for other YTH domain containing proteins (*Patil et al., 2018*).

## Pho92 promotes decay of m6A modified transcripts

Proteins of the YTHDF family, to which Pho92 belongs, promote mRNA decay but have also been shown to impact translation efficiency (*Wang et al., 2014*; *Wang et al., 2015*; *Li et al., 2017*; *Zaccara and Jaffrey, 2020*). To determine whether Pho92 plays a possible role in mRNA decay, we reasoned that if Pho92 is indeed important for the decay of m6A-modified transcripts, *pho92Δ*cells should also display higher m6A levels. Hence, we determined the m6A levels by LC-MS of cells entering meiosis (4 h SPO) and compared the wild type to *pho92Δ*. We found that mRNAs isolated from *pho92Δ*cells displayed a marked increase in m6A over adenosine (A) levels (*Figure 5A*). As expected, *ime4Δ* cells showed no detectable m6A levels.

We also performed RNA-seq (*pho92Δ* vs WT) of cells staged prior to meiosis when Pho92 is weakly expressed and m6A is relatively low (0 hr SPO) or undergoing early meiosis when Pho92 is strongly induced and m6A is high (4 hr SPO). We examined the Pho92 bound Ime4-dependent and Ime4-independent transcripts identified by Pho92 iCLIP (*Figure 5B*). At 0 hour in SPO, we observed little difference between *pho92Δ*and the WT (*Figure 5B*, *Figure 5—figure supplement 1A*). At 4 hr in SPO, we observed a notable increase in expression for Pho92 bound transcripts that were Ime4-dependent, but not for the Ime4-independent transcripts (*Figure 5B*, *Figure 5—figure supplement 1A*). Out of 295 transcripts significantly up-regulated in *pho92Δ* vs WT (4 hours SPO),95 were Pho92 targets as determined by iCLIP (*Figure 5—figure supplement 1A*). Moreover, transcripts significantly upregulated in *pho92Δ*at 4 hours in SPO showed greater binding of Pho92 at 3'ends compared to genes that either did not change in expression or were downregulated (*Figure 5C*). We observed a similar link between Pho92 Ime4-dependent transcripts and increased expression in *ime4Δ* cells during entry into meiosis (*Figure 5—figure supplement 1B, C, D*). For example, 212 out of 505 Pho92 iCLIP targets were significantly upregulated in *ime4Δ* vs WT (4 hr SPO) (*Figure 5—figure supplement 1B*). We conclude that Pho92 limits the accumulation of m6A-modified transcripts.

Measuring the rate of mRNA decay of m6A transcripts is not trivial, foremost because m6A transcripts can be masked by presence of non-m6A transcripts within a pool of RNAs (*Garcia-Campos et al., 2019*). Additionally, measuring mRNA decay in sporulating cells using metabolic labelling approaches (e.g. SLAM-seq) is problematic, due to slow uptake of exogenous labelled nucleotides under these conditions (*Herzog et al., 2017*). Hence, we decided to measure mRNA decay rates using two different approaches. First, we shut-off the transcription of early meiotic genes by switching to nutrient rich conditions (YPD), exiting cells from meiosis and returning cells to growth (RTG). Subsequently we measured the mRNA half-life ($t_{1/2}$) in wild-type and *pho92Δ* cells of specific target transcripts (*GUT2*, *HOP1*, *IME1*). Consistent with previous work, we found that *GUT2*, *HOP1* and *IME1* mRNA levels rapidly decreased with estimated mRNA $t_{1/2}$ between 2 and 4 min (*Figure 5D*; *Chia et al., 2021*). In *pho92Δ* cells, the mRNA $t_{1/2}$ of these transcripts were noticeable longer (approximated 1.6-fold) between 3 and 6 min. Additionally, the *IME2* transcript, which showed no detectable Pho92 binding (*Figure 2B*), showed marginal difference in mRNA $t_{1/2}$ (1.2-fold, 3.5 min for WT vs 4.3 min for *pho92Δ*; *Figure 5—figure supplement 1E*).

We adopted a second approach to measure decay of m6A-transcripts. We reasoned that if m6A modified transcripts are less stable compared to non-m6A mRNAs, m6A is expected to decay faster after blocking mRNA synthesis. We treated cells with thiolutin to block transcription, and subsequently determined m6A levels by m6A-ELISA or LC-MS in wild-type and *pho92Δ* cells during early meiosis (4 hr SPO) (*Figure 5E*, *Figure 5—figure supplement 2A*). Overall m6A levels declined after thiolutin treatment, indicating that m6A modified mRNAs are less stable than unmodified mRNAs. In *pho92Δ*-cells however, m6A levels did not decline after thiolutin treatment (*Figure 5E*). Similarly, *PHO92*[W177A]

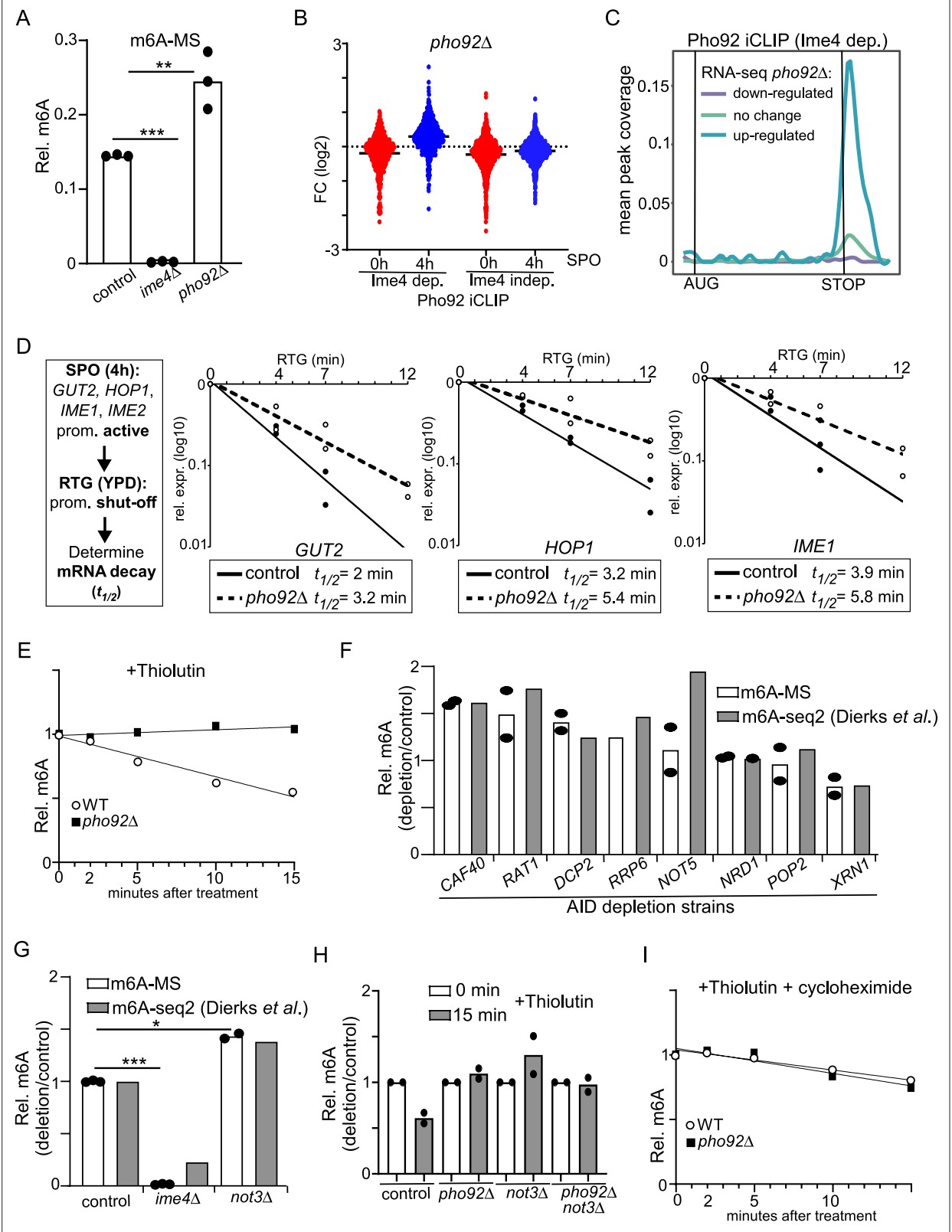

**Figure 5.** Pho92 and CCR4-NOT promotes decay of m6A marked transcripts. (**A**) m6A-MS for control, *ime4Δ*, and *pho92Δ* (FW4911, FW6060, and FW6997). Samples were taken at 6 h in SPO. mRNA was isolated using oligodT paramagnetic beads. Samples were digested, and m6A levels of A were determined by LC-MS. **p<0.01 and ***p<0.001, compared to WT control on a two-way ANOVA followed by a Fisher's least significant difference (LSD) test. n=3 biological repeats. (**B**) Differential analysis of *pho92Δ* vs WT for transcripts with Ime4-dependent and Ime4-independent Pho92 binding sites.

*Figure 5 continued on next page*

*Figure 5 continued*

Samples were taken at 0 and 4 hours in SPO for WT and *pho92Δ* cells (FW1511 and FW3528). RNA-seq was performed. Displayed are transcripts that are bound by Pho92 in an Ime4-dependent and independent manner as determined by iCLIP. Differential expression between 0 hr in SPO for *pho92Δ* vs WT are displayed (red), and 4 hr in SPO *pho92Δ* vs WT are displayed (blue). (C) Metagene analysis plotting Ime4-dependent Pho92 binding sites on genes that were either significantly upregulated, down regulated, or unchanged in 4 hr in SPO *pho92Δ* vs WT RNA-Seq. The y-axis represents the proportion of transcripts in that RNA-seq category with a binding site overlapping a given x coordinate. (D) mRNA decay measurements after transcriptional shut-off by returning cells to growth (YPD) for specific meiotic mRNAs (*GUT2, HOP1, and IME1*). For the analysis samples were taken at 4 hours in SPO (0 min timepoint), and 4, 7, and 12 min after return of cells to growth. mRNA expression was determined by RT-PCR. Samples were normalized by *ACT1*. Indicated are the estimated mRNA half-lives ($t_{1/2}$) based on n=2 biological repeats. (E) m6A-ELISA in WT and *pho92Δ* cells (FW1511 and FW3528) after blocking transcription using thiolutin. Cells were grown and induced to enter meiosis. Cells were treated with thiolutin, and samples were taken at the indicated time points. Relative m6A levels were determined by m6A-ELISA. (F) Relative m6A levels in depletion mutants of various decay pathways determined by m6A-MS and m6A-seq2 (FW5958, FW6080, FW6048, FW5880, FW 6070, FW6043, FW5956, FW5964). In short, cells were grown and induced to enter meiosis. Each decay mutant bearing auxin-induced depletion alleles (AID) was depleted by adding auxin at 4 hr in SPO, and samples were collected at 6 hr in SPO. m6A-MS (n=2 biological repeats) and m6A-seq2 data were normalized to a control strain harbouring the *TIR* ligase. m6A-seq2 data were obtained from *Dierks et al., 2021*. (G) Similar analysis as E, except that the signals for *ime4Δ* and *not3Δ* cells are shown (control, FW4911, FW6060 and FW6093). *p<0.05 and ***p<0.001, for m6A-MS compared to WT control on a two-way ANOVA followed by a Fisher's least significant difference (LSD) test. WT and *ime4Δ* data are the same as in A. At least n=2 biological repeats were performed. (H) m6A-ELISA comparing WT, *pho92Δ* and *not3Δ* single and double mutants (FW1511, FW3528, FW6090, and FW6179) before and after blocking transcription with thiolutin for 15 min. Cells were grown to and induced to enter meiosis, treated with thiolutin, and samples were taken at 15 min after treatment. Relative m6A levels were determined by m6A-ELISA. n=2 biological repeats. (I) Similar analysis as E, but cells were treated with both thiolutin and cycloheximide.

The online version of this article includes the following figure supplement(s) for figure 5:

**Figure supplement 1.** Pho92 and CCR4-NOT promotes decay of m6A marked transcripts.

**Figure supplement 2.** Pho92 and CCR4-NOT promotes decay of m6A marked transcripts.

mutation, which disrupts Pho92 binding to m6A, also showed no decline of m6A-mRNA levels after thiolutin treatment (*Figure 5—figure supplement 2B*). Taken together, these data support a role for Pho92 in promoting the decay of m6A modified mRNAs.

## CCR4-NOT mediates m6A decay

Pho92 itself has no known enzymatic activity that would drive the decay of transcripts. To identify protein complexes involved in the decay of m6A modified transcripts via Pho92, we depleted or deleted various genes involved in mRNA decay and measured the effect on m6A levels via LC-MS in meiosis (6 hr SPO) (*Figure 5F and G*). Essential components were depleted using induction of the AID system at 4 hr in SPO and m6A levels were determined at 6 hr SPO (2 hr after depletion). We also compared the LC-MS data with m6A-seq2 data, a technique that relies on multiplexed m6A-immunoprecipitation of barcoded and pooled samples, of the same depletion alleles (*Dierks et al., 2021*). We found that *not3Δ* displayed an increase in m6A levels detected by LC-MS and m6A-seq2 (*Figure 5G*). Moreover, depletion of Caf40 and Rat1 (*CAF40-AID* and *RAT1-AID*) showed a consistent increase in m6A levels using both LC-MS and m6A-seq2 (*Figure 5F*). In contrast depletion of Xrn1 (*XRN1-AID*) had the opposite effect and showed reduced m6A levels. Our data suggest that the various mRNA decay pathways can have opposing effects on m6A levels during yeast meiosis.

Both Caf40 and Not3 are part of the CCR4-NOT complex the major mRNA deadenylation complex in eukaryotes (*Collart, 2016*). Various reports have shown that YTH reader proteins facilitate recruitment of the CCR4-NOT complex to target RNAs (*Kang et al., 2014*; *Du et al., 2016*). We determined whether the CCR4-NOT complex had a similar effect on the decay of m6A modified transcripts as Pho92. We blocked transcription in *not3Δ* diploid cells entering meiosis, and measured m6A levels. We found that like Pho92, the decay of m6A-modifed transcripts was reduced in *not3Δ* diploid cells when compared to wild-type cells (*Figure 5H*). Thus, CCR4-NOT facilitates the decay of m6A modified transcripts. Given that CCR4-NOT and Pho92 interact with each other, our data suggest that Pho92 promotes decay of its target transcripts via the CCR4-NOT complex.

## Pho92 and CCR4-NOT mediated m6A decay requires translation

Among Pho92 interactors several proteins involved in translation and ribosome biogenesis were identified (*Figure 4A*). In addition to the vast literature on the intimate connection between mRNA decay rate and translation elongation, a physical link between CCR4-NOT and ribosomes has also been established, implicating CCR4-NOT complex directly in translation (*Buschauer et al., 2020*). This

prompted us to examine whether the decay of m6A modified transcripts relied on active translation. In addition to blocking transcription, we also inhibited translation by treating cells with cycloheximide. Blocking transcription and translation alleviated the differences in decay rates between *pho92Δ*and wild-type cells, suggesting that the decay of m6A-modified transcripts is certainly dependent on translation (*Figure 5I*, *Figure 5—figure supplement 2A*). Noteworthily, upon cycloheximide treatment in both *pho92Δ*and wild-type cells a small decline in m6A levels was detected. A possible explanation is that levels of the m6A writer machinery is reduced after blocking translation. Similarly, blocking translation and transcription alleviated the differences between *not3Δ*and wild type (*Figure 5—figure supplement 2C*). We conclude that Pho92 stimulates the decay of transcripts of m6A-modified transcripts in a translation-dependent manner and, at least in part, via the CCR4-NOT complex.

## Pho92 interacts with ribosomes

The observation that Pho92 stimulates decay of m6A marked transcripts in a translation-dependent manner, suggests that Pho92 functions at ribosomes. To investigate whether Pho92 indeed associates with ribosomes, we prepared cell lysates enriched for ribosomes from cells expressing Pho92 in cycling cells by pelleting extracts with high-speed centrifugation. We observed that majority of Pho92 sedimented together with ribosomes in the pellet (*Figure 6A*, *Figure 6—figure supplement 1A*). Additionally, we performed polysome profiling to determine whether Pho92 associates with actively translating ribosomes in diploid cells undergoing early meiosis (*Figure 6—figure supplement 1B*). We found that Pho92 associates with polysome fractions as well as monosome fractions (*Figure 6B*), and hence conclude that Pho92 associates with ribosomes including translating ribosomes.

## Pho92 target transcripts are linked to increased translation efficiency

Our observation that Pho92 associates with ribosomes presented the possibility that Pho92 also regulates translation. We assessed the translation efficiency of Pho92-bound transcripts by performing RNA-seq on RNA isolated from polysome fractions, known as polysome profiling (*King and Gerber, 2016*). We found that the Pho92 iCLIP targets were similarly upregulated in the polysome fractions compared to total RNA fraction (*pho92Δ* vs WT, 4 hr SPO), suggesting that loading of ribosomes to Pho92 target mRNAs was not directly affected in *pho92Δ*cells (*Figure 6C*). To obtain a better resolution view of translation throughout meiosis, we assessed a ribosome footprinting dataset and compared the translation efficiency of Pho92 targets (*Brar et al., 2012*). Transcripts bound by Pho92 such as *BDF2* (*Figure 1I*) displayed increased ribosome footprints specifically during pre-meiotic DNA replication and recombination (*Figure 6—figure supplement 1C*). Meta-analysis revealed that the median translation efficiency (TE) was higher for Pho92 target transcripts (Pho92 iCLIP) and m6A modified transcripts (miCLIP) during premeiotic DNA replication, and recombination compared to a control set of transcripts (*Figure 6D*). These stages of meiosis also match the stages when m6A is most enriched (*Schwartz et al., 2013*). Thus, Pho92-bound transcripts have a higher translation efficiency during early meiosis and increased decay rates, which is unusual as transcripts with increased decay rates typically have a lower translation efficiency (*Presnyak et al., 2015*). Taken together, these data suggest that Pho92 function in addition to regulating decay, is linked to some aspect of translation.

## Pho92 stimulates protein synthesis of its targets

To determine how Pho92 and m6A control protein synthesis more directly, we performed quantitative proteomics in WT and *pho92Δ*cells during early meiosis and compared the data to RNA-seq (*Figure 6E*, *Figure 6—figure supplement 1D*). We reasoned that if Pho92 stimulates protein synthesis then in *pho92Δ*cells the protein over RNA ratio must be lower. We examined a set of genes that showed Ime4-dependent binding of Pho92 and a control set of genes. To highlight differences, we grouped the RNA-seq data (WT vs *pho92Δ*) into three categories: genes that were down-regulated (1), showed little change (2) or were up-regulated (3). For Pho92-bound transcripts, protein and RNA changes were not notably different for the group of genes that were either down-regulated (1) or showed little change in the RNA-seq (2) (*pho92Δ* vs WT) (*Figure 6E*, *Figure 6—figure supplement 1D*). However, genes that were up-regulated (3) in the RNA-seq (*pho92Δ* vs WT), were not upregulated for the protein products (*Figure 6E*, *Figure 6—figure supplement 1D*). In contrast, the control set of genes that were up-regulated in the RNA-seq (*pho92Δ* vs WT), showed an increase in protein levels (*Figure 6E*, *Figure 6—figure supplement*

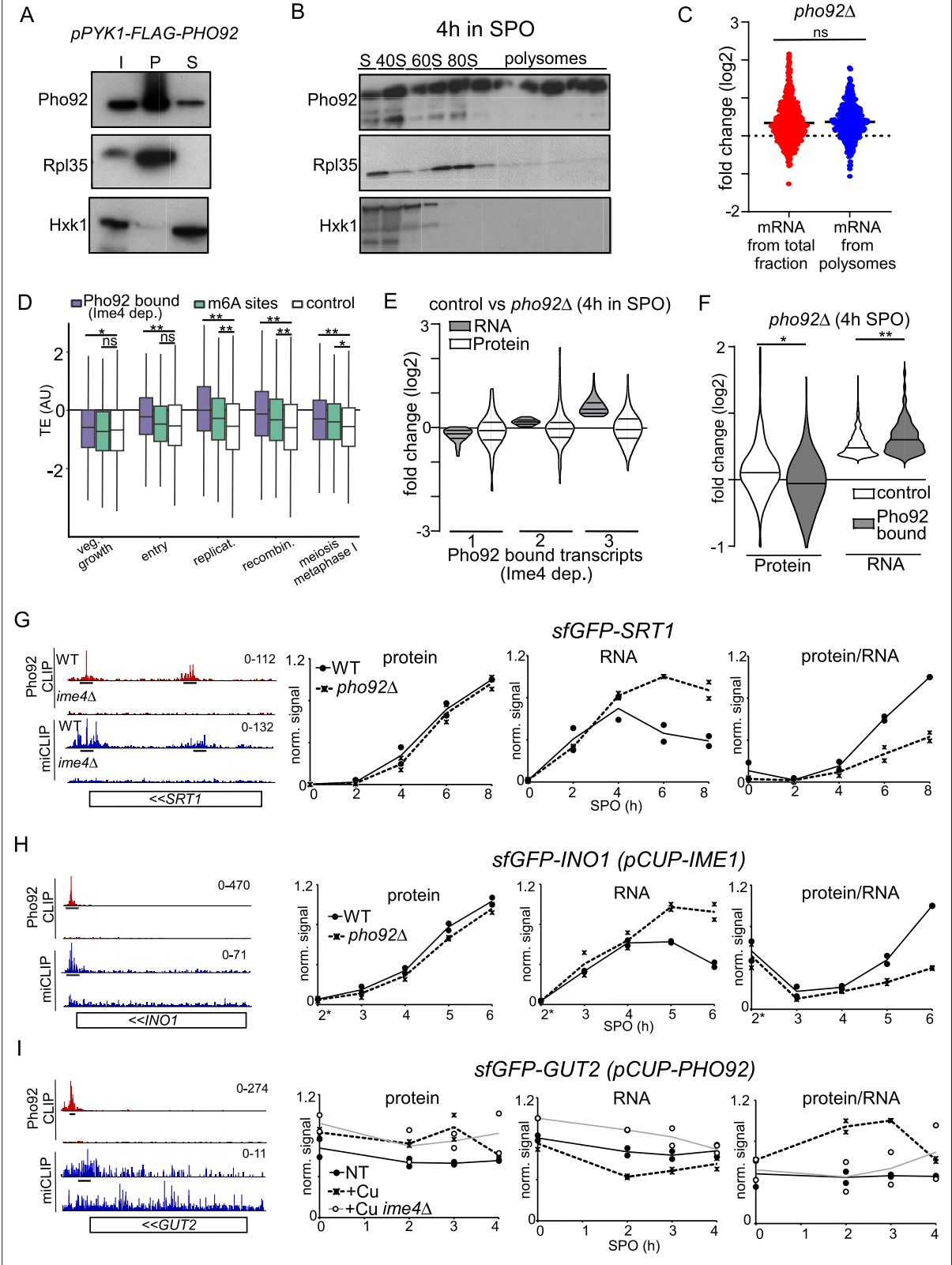

**Figure 6.** Pho92 interacts with polysomes and controls protein expression. (**A**) Sucrose cushion analysis of cells expressing *pPYK1*-FLAG-Pho92 (FW 8732) were grown in rich medium. Immunoblots of total (T) pellet/ribosomal (P) and soluble (S) fractions are shown for FLAG-Pho92, hexokinase (Hxk1) and Rpl35 (ribosomal subunit of 60 S) proteins. (**B**) Polysome fractionation and western blot of Pho92. Diploid cells harbouring Pho92 tagged with V5 (FW4478). Cells were induced to enter meiosis and at 4 hr in SPO samples were taken. The small (40 S), large (60 S), both (80 S) subunits and polysomes

*Figure 6 continued on next page*

*Figure 6 continued*

are highlighted from the polysome traces. Protein extracts from fractions were probed for Pho92-V5 by immunoblotting. As controls membranes were probed for Rpl35 and Hxk1. (**C**) Polyribo-seq analysis was performed 4 hr SPO *pho92Δ* vs WT (FW1511 and FW3528). Ribosomal fractionation was performed to isolate mRNAs bound to polysomes. Polyribo-seq was performed and the data was compared to RNA-seq from total fraction. Displayed are transcripts that are bound by Pho92 in an Ime4-dependent manner as determined by iCLIP. Differential expression between *pho92Δ* vs WT are displayed for total mRNA from total fraction (red), and polysome fraction (blue). Non-significant (ns) compared to WT control on a Welsch's t-test. (**D**) Analysis of translation efficiency (TE) using **Brar et al., 2012** dataset. We assessed the TE for transcripts harbouring Ime4-dependent Pho92 binding sites, m6A sites and a control set of transcripts comprised of the rest of expressed mRNAs at 4 hr in SPO. Boxes show the median and interquartile range, extending lines show first and fourth quartiles. *p<0.005 and **p<0.001 compared to control using Welsch's t-test. (**E**) Violin plots describing RNA-seq and proteome data from comparing *pho92Δ*(FW3528) to control (FW1511). In short, cells were induced to enter meiosis. Samples for RNA-seq and whole proteome analysis were taken at 4 hr in SPO for WT and *pho92Δ*cells. Transcripts with Ime4-dependent Pho92 binding sites and with a signal in whole proteome quantification were used for the analysis. The RNA-seq data divided in the three groups (1–3), group 1 represents genes with the reduced RNA-seq signal, group 2 little change, group 3 upregulated in *pho92Δ* RNA-seq. The corresponding signal from the whole protein data is displayed. (**F**) Similar analysis as E, except that the control entailed a group of genes up-regulated in the RNA-seq that were not bound by Pho92 were used for the analysis. *p<0.05 and **p<0.01 compared to control on a Welsch's t-test. (**G–I**) *SRT1, GUT2* and *INO1* transcripts bound by Pho92 and marked with m6A. Shown are the data tracks for Pho92 CLIP for WT (FW4472) and *ime4Δ* (FW4505) cells, and miCLIP for WT (FW1511) and *ime4Δ* (FW725). Underlined are the binding sites identified in the analysis. Tracks are crosslink per million normalised stranded bigWigs viewed in IGV. Relative protein, RNA, and protein over RNA ratios. WT and *pho92Δ* cells were grown to enter meiosis and samples from time points indicated. For the analysis *SRT1, INO1, and GUT2* were tagged seamlessly with sfGFP at the amino terminus. Protein expression was determined by western blotting. The relative signal is displayed with max signal for each biological repeat (n=2) scaled to one. RNA levels were determined by RT-qPCR. The relative signal was computed by setting the maximum signal for each time course experiment (which included WT and *pho92Δ*) repeat to one. Right panel. The ratio of protein and RNA signals. The relative signal was computed by setting the maximum signal for each time course experiment (which included WT and *pho92Δ*) repeat to 1. For *SRT1* analysis, cells were induced to enter meiosis at the indicated time points (FW9949 and FW9950). For *INO1* analysis, meiosis was induced using *pCUP-IME1* synchronization system (FW9746 and FW9747). For *GUT2* analysis was performed in the presence or absence of Pho92 expression from the *CUP1* promoter (FW 10438 and *ime4Δ*, FW10441).

The online version of this article includes the following source data and figure supplement(s) for figure 6:

**Source data 1.** Raw scans of *Figure 6A*.

**Source data 2.** Raw scans of *Figure 6A*.

**Source data 3.** Raw scans of *Figure 6A*.

**Source data 4.** Raw scans of *Figure 6B*.

**Source data 5.** Raw scans of *Figure 6B*.

**Source data 6.** Raw scans of *Figure 6B*.

**Figure supplement 1.** Pho92 interacts with polysomes and controls protein expression.

**Figure supplement 1—source data 1.** Raw scans of *Figure 6—figure supplement 1A*.

**Figure supplement 1—source data 2.** Raw scans of *Figure 6—figure supplement 1A*.

**Figure supplement 1—source data 3.** Raw scans of *Figure 6—figure supplement 1A*.

**Figure supplement 1—source data 4.** Licor Odyssey multi-channel scan of western blot probed for SfGFP-Srt1 and Hxk1 in *Figure 6—figure supplement 1E*.

**Figure supplement 1—source data 5.** Licor Odyssey multi-channel scan of western blot probed for SfGFP-Ino1 and Hxk1 in *Figure 6—figure supplement 1E*.

**Figure supplement 1—source data 6.** Licor Odyssey multi-channel scan of western blot probed for SfGFP-Gut2 and Hxk1 (reprobed for HA-Pho92) in *Figure 6—figure supplement 1E*.

**Figure supplement 1—source data 7.** Licor Odyssey multi-channel scan of western blot probed for SfGFP-Gut2 and Hxk1 in *Figure 6—figure supplement 1F*.

**Figure supplement 1—source data 8.** Licor Odyssey multi-channel scan of western blot probed for RGT1-V5 and Hxk1 in *Figure 6—figure supplement 1G*.

**Figure supplement 1—source data 9.** Licor Odyssey multi-channel scan of western blot probed for BDF2-V5 and Hxk1 in *Figure 6—figure supplement 1H*.

*1D*). We noticed that the increase in the RNA-seq (*pho92Δ* vs WT) of this control group was less compared to Pho92-bound transcripts. This prompted us to generate a second control set of genes that showed approximately equal increase in the RNA-seq (*pho92Δ* vs WT) but no binding of Pho92. This control group of genes also showed an increase in protein levels following the RNA-seq (*pho92Δ* vs WT), while the Pho92-bound transcripts displayed a small decrease in protein level

despite the RNA levels were more increased for this set of transcripts (*pho92Δ* vs WT) (**Figure 6F**). We conclude that Pho92-bound transcripts with increased RNA levels were not followed by an increase in protein levels in *pho92Δ* cells, supporting the idea that Pho92 stimulates protein synthesis of its target mRNAs.

To confirm if the aforementioned trend of discordant protein and mRNA levels can also be observed for individual genes, we selected several transcripts with Ime4-dependent Pho92 binding sites to test further (**Figure 6G–I**, **Figure 6—figure supplement 1E–H**). We tagged *INO1*, *SRT1*, *GUT2* at the amino-terminus seamlessly with sfGFP. Also, we tagged *RGT1* and *BDF2* at carboxy-terminus with the V5 tag. Subsequently, we performed time courses covering early meiosis from cells either directly shifted to starvation (SPO medium) or used *pCUP1-IME1* to induce meiosis in a highly synchronous manner. For each time course experiment, we determined the relative protein levels, RNA levels, and computed the protein over RNA ratio. Consistent with the proteomics and RNA-seq data, we found that *SRT1*, *INO1*, *GUT2*, *RGT1*, and *BDF2* displayed a decrease in protein over RNA levels in *pho92Δ* cells over several time points (**Figure 6G–I**, **Figure 6—figure supplement 1E–H**). *SRT1* showed increased RNA levels in *pho92Δ* cells compare to the control, while protein levels were higher in control cells (**Figure 6G**). Like *SRT1*, albeit less, the protein over RNA ratios decreased for *GUT2* and *RGT1* in *pho92Δ* cells (**Figure 6—figure supplement 1F and G**). For the analysis where *pCUP-IME1* was induced for synchronization, we observed that the protein over RNA ratios of *INO1* and *BDF2* were decreased in *pho92Δ* cells compared to the control at later time points (**Figure 6H**, **Figure 6—figure supplement 1E, H**).

Vice versa, we examined whether overexpression of Pho92 can have the opposite effect on protein over mRNA ratios. We expressed Pho92 to high levels from the *CUP1* promoter (*pCUP-PHO92*) and measured protein and RNA levels for *GUT2* (**Figure 6I**, **Figure 6—figure supplement 1E**). In the presence of high levels of Pho92 (*pCUP-PHO92* + Cu) we observed an increase in Gut2 protein levels while RNA levels decreased, and consequently a substantial increase in protein over RNA ratio (**Figure 6I**, **Figure 6—figure supplement 1E**). The effect was dependent on the presence of Ime4. Taken together, we conclude that Pho92 associates with ribosomes to stimulate mRNA decay and protein synthesis of m6A modified transcripts.

## Discussion

m6A modified transcripts are abundant during early yeast meiosis, yet the function of the m6A mark remains elusive. Here we show that the YTH domain containing protein Pho92/Mrb1 is a developmentally regulated m6A reader in yeast that is critical for the onset of meiosis and fitness of gametes. Importantly, we provide evidence that Pho92 co-transcriptionally associates with mRNAs to direct m6A-modifed transcripts for their translation to decay fate. Our study further shows that Pho92 couples protein synthesis and mRNA decay for m6A modified transcripts (**Figure 7**). Our work in yeast meiosis establishes a possible link between the previously described decay and translation functions of YTH reader proteins and m6A-modified mRNAs.

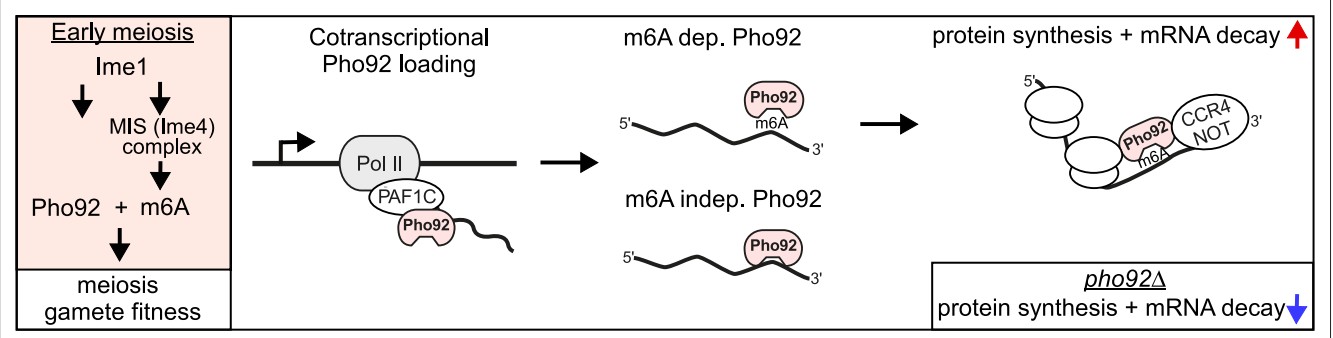

**Figure 7.** Model for role of Pho92 in early meiosis. Pho92 and MIS complex (Ime4) expression are induced by Ime1. Subsequently, Pho92 is loaded to mRNAs during transcription via Paf1c, and Pho92 promotes the protein synthesis to mRNA decay fate of m6A modified transcripts involving CCR4-NOT.

## Pho92 is key to meiotic fitness

During yeast meiosis, which is induced by severe nutrient starvation, diploid cells are triggered to undergo inarguably the most complex cell differentiation program of the yeast life cycle (*Neiman, 2011*; *van Werven and Amon, 2011*). Master transcription factor Ime1, which drives the transcription of genes in early meiosis, likely directly regulates *PHO92* transcription. Thus, in yeast the m6A writer and reader machinery (Pho92 and the MIS complex) are specifically induced during early meiosis (*Figure 7*). Interestingly, Pho92 also regulates the stability of the *PHO4* transcript in a phosphate-dependent manner, suggesting that Pho92 may also play an additional role outside meiosis when there is no m6A (*Kang et al., 2014*).

Tight regulation of gene expression is critically important to ensure no cellular resources are wasted during meiosis. With this view, the role of Pho92 in coupling translation efficacy to mRNA decay provides an elegant strategy to facilitate increased protein synthesis and subsequent decay of mRNAs important for meiosis. One other possibility is that Pho92 is important for decay of mitotic transcripts, so that cells can enter gametogenesis, or decay of early meiotic transcripts, so that cells enter the subsequent stages of gametogenesis. If this is the case, one expects to find the m6A modified transcripts to be confined to a defined stage corresponding to either the exit from mitosis or entry into meiotic divisions. However, we observe no enrichment for genes linked to mitosis in the Pho92 iCLIP data, and m6A levels are spread over several stages during early meiosis (*Schwartz et al., 2013*). Rather, our data suggests that Pho92 increases the translation efficacy of m6A-modified transcripts during critical stages of the early meiotic program when errors in meiotic chromosome segregation are fatal to the survival of gametes. Indeed, cells lacking Pho92 exhibit a delay in the onset of meiotic divisions and reduced gamete fitness.

While Pho92 deletion mutants exhibit a delay in meiosis, an Ime4 deletion mutant has more severe phenotype (*Figure 3*). Apart from the function that m6A has to mediate the Pho92 interaction, it is plausible that m6A has additional roles. However, it is worth noting that a catalytic dead mutant of Ime4 has a less severe phenotype in meiosis, which lead to the conclusion that Ime4 has noncatalytic function during meiosis (*Agarwala et al., 2012*). Therefore, we propose that the implications of the m6A modification is largely manifested through Pho92.

## Mechanism of co-transcriptional loading of Pho92

Our data suggests that Pho92 interacts with target transcripts during transcription via transcription elongator Paf1C (*Figure 7*). First, we found that Pho92 can interact with Paf1C, and that this interaction is critical for the localisation of Pho92 to the nucleus. Interestingly, the Leo1 subunit of Paf1C is also known to interact with nascent RNA, and thus forms a logical platform to facilitate the interaction between Pho92 and nascent RNA (*Dermody and Buratowski, 2010*). Pho92 binding sites show striking 3' end enrichment regardless of whether binding is Ime4-independent or at m6A sites, which would suggest that the positioning is not entirely dependent on m6A. Pho92 accumulates at chromatin over the course of gene transcription, in a manner that is dependent on Paf1C. This would suggest that transcription or chromatin factors contribute towards Pho92 positioning on transcripts, which may explain why Ime4-independent Pho92 binding does not need a strong RNA sequence context. A simple mechanism could be that m6A locks Pho92 binding to RNA, which in turn allows the Pho92-mRNAs to be sent to translating ribosomes (*Figure 7*). It is likely that without m6A locking Pho92 in place, the protein-RNA interaction is less stable. Another possibility is that Pho92 has a separate regulatory function in the nucleus involving chromatin and transcription as has been reported for other YTH domain containing proteins (*Patil et al., 2018*).

Transcription and chromatin-mediated loading of RNA binding proteins and m6A machinery to control RNA fates is not only limited to Pho92. The co-transcriptional loading of RNA binding proteins has been reported for other RNA binding proteins to control RNA export (*Shahbabian et al., 2014*; *Fischl et al., 2017*; *Viphakone et al., 2019*). There is also evidence that the promoter sequences influence the stability of RNA (*Trcek et al., 2011*). With regard to m6A, there is proof that m6A is deposited early in RNA synthesis either co-transcriptionally or via chromatin (*Ke et al., 2017*; *Huang et al., 2019*). YTH proteins can act in the nucleus to facilitate the decay of nuclear m6A-modified transcripts and thereby control chromatin states (*Liu et al., 2020*). In *S. pombe*, the YTH-RNA-binding protein Mmi1 is also reported to be co-transcriptionally recruited to 'decay-promoting' introns bearing sequence elements known as 'determinants of selective removal' (DSRs), which are enriched

in meiotic mRNAs (*Kilchert et al., 2015*). Contrary to other known YTH-containing domain proteins including Pho92, Mmi1 is incapable of binding to the m6A consensus motif and the Mmi1 role is in fact reported to be suppression of activation of the meiotic programme during mitotic growth, which is in contrast to Pho92's role in meiosis in *S. cerevisiae* (*Harigaya et al., 2006*; *Wang et al., 2016*; *Shichino et al., 2018*).

## The role of Pho92 in the translation to decay fate

Our analysis of Pho92 shows that the destiny of m6A-marked transcripts in translation and mRNA decay are linked. On one hand, we have experimental evidence showing Pho92 is important for efficient protein production. Deletion of Pho92 leads to a reduction in protein over RNA ratios, suggesting translational efficacy is lowered. Additionally, Pho92-bound transcripts showed increased translation efficiency compared to a control set (*Figure 6*). On the other hand, Pho92 stimulates the decay of m6A-modified transcripts. How does Pho92 promote decay as well as protein synthesis of m6A-modified transcripts? While it is not impossible that the translation and decay functions of Pho92 are exerted via distinct mechanisms, the likelihood of these functions being linked is more plausible. Firstly, the decay of m6A marked transcripts bound by Pho92 occurred in a translation dependent manner. Secondly, we found that Pho92 associates with actively translating ribosomes and is directly linked to increased translation efficacy. Thirdly, our analysis of decay mutants revealed that the CCR4-NOT complex facilitates the decay of m6A-marked transcripts (*Figure 5*). Specifically, depleting the Not3 and Caf40 subunits resulted in increased m6A levels. CCR4-NOT is the major deadenylase complex in cells, which acts at ribosomes and facilitates co-translational decay (*Collart, 2016*). Though we did not establish a direct interaction between Pho92 and the CCR4-NOT complex in this study, others have shown that Pho92 interacts with the Pop2 subunit of CCR4-NOT (*Kang et al., 2014*).

During the review of our manuscript, a related manuscript was published with several overlapping findings to our work (*Scutenaire et al., 2022*). In this work, the authors also showed that Pho92 associates with m6A marked mRNAs, promotes progression of meiosis, and stimulates the repression of its target mRNAs. The authors propose that timely repression of specific meiotic mRNAs is key for the transition into meiosis. Based on our more in-depth analysis, we favour a model where Pho92 promotes translation and subsequent decay of meiotic transcripts (see discussion above). In our view is unlikely that the relatively small differences in mRNA decay rates (only a few minutes difference) between wild-type and *pho92Δ*cells can directly explain the delay in meiotic progression of a few hours as observed in *pho92Δ*cells. How Pho92 directly controls progression into meiosis remains to be further investigated.

In mammalian cells, YTHDF1 and YTHDF2 have been implicated in translation and decay, respectively (*Wang et al., 2015*; *Zaccara and Jaffrey, 2020*). Also YTHDC2 has been assigned both functions in decay and translation (*Hsu et al., 2017*; *Kretschmer et al., 2018*; *Mao et al., 2019*). Thus, opposite effects have been reported for YTH reader proteins. YTHDF2, the likely orthologue of Pho92, has been shown to be important for oocyte development, where it is important for decay of maternal RNAs (*Ivanova et al., 2017*). YTHDF2 associates with CCR4-NOT in mammalian cells (*Du et al., 2016*). Thus, the link between CCR4-NOT, YTH domain-containing proteins and m6A is likely conserved in yeast and mammalian gametogenesis. Interestingly, the translation function of YTHDF1 has been disputed, and it has been proposed that YTHDF proteins act redundantly to regulate decay (*Zaccara and Jaffrey, 2020*). Contradicting this, a significant enrichment of m6A in yeast ribosomal fractions and increased methylation of the most efficiently translated transcripts in the early stages of meiosis has been reported (*Bodi et al., 2015*). Perhaps, the decay functions of Pho92 and YTHDF1 at ribosomes facilitates translational efficiency and as consequence decay or *vice versa*. The reason that mammalian studies have failed to conclude that decay and translation functions may go hand in hand is perhaps because mammalian studies of YTH proteins have typically focused on cell lines under steady state conditions (*Zaccara and Jaffrey, 2020*). In contrast, yeast meiosis is a dynamic process requiring extremely quick turn over of cell RNA and protein states, which is made possible by tightly coupled RNA decay and translation.

More and more studies show that an intimate link between translation and decay exists (*Pelechano et al., 2015*; *Presnyak et al., 2015*). Slowing down translation elongation has been linked to slowing the decay of transcripts (*Chan et al., 2018*). Perhaps, Pho92 directly promotes translation elongation,

which in turn leads to faster decay via the CCR4-NOT complex. As such, there is evidence that CCR4-NOT complex, apart from its direct function in translation coupled decay, monitors translating ribosomes for codon optimality (*Buschauer et al., 2020*). Another intriguing possibility is that m6A and Pho92, together with CCR4-NOT are part of an mRNA quality control mechanism to limit aberrant protein synthesis and protein misfolding, while at the same time increase productive translation. In line with this idea is that YTHDC2 protein facilitates interactions between m6A, ribosomes, and decay factors to control stability and translation of the mRNAs, and together possibly constitutes an mRNA to protein quality control mechanism (*Kretschmer et al., 2018*; *Inada, 2020*). How m6A, Pho92 and CCR4-NOT together promote translation and decay and how this is linked to a quality control for mRNAs and aberrant translation remains subject for further investigation.

YTH reader proteins are conserved from yeast to humans and play critical roles in developmental programs and disease pathogenesis (*Patil et al., 2018*). YTHDF1 has been shown to be involved in tumour cells immune evasion, while YTHDF2 promotes tumour cell proliferation (*Chen et al., 2017*; *Han et al., 2019*). The involvement of YTHDC1 in splicing has been associated with the incorrect processing of BRCA2, and YTHDC2 was found to contribute to metastasis (*Hirschfeld et al., 2014*; *Tanabe et al., 2016*). Additionally, YTHDF2 is a potential therapeutic target in acute myeloid leukaemia treatment (*Mapperley et al., 2021*). Understanding the molecular mechanisms of Pho92 in translation and decay in yeast may therefore reveal novel insights into human health and disease pathogenesis.

## Materials and methods
### Plasmids and yeast strains
Yeast strains used in this study were derived either from the sporulation proficient SK1, or poor sporulation S288C or A364A backgrounds. All experiments were carried out in diploid cells apart from the experiments using strains expressing N-terminal Flag tagged *PHO92* from *PYK1* or *CYC1* promoters.

GST-tagged Pho92 full-length, truncations NTDΔ (containing residues 141–306) and YTHΔ (containing residues 1–140) and GST-tagged Gis2 were generated by Gibson cloning using yeast genomic DNA as template. *CUP1-HA-PHO92*-WT and *CUP1-HA-PHO92-W177A* were generated by Gibson cloning into pNH604 vector and integrated in the *TRP* locus. *IME4^WT* and *IME4^CD* were cloned by Gibson cloning into pNH604 vector and *IME4^CD* mutation was previously described (*Clancy et al., 2002*). Plasmids and oligonucleotide sequences are listed in the *Supplementary file 2* and *Supplementary file 3*, respectively.

Gene deletions and epitope tagging were achieved using the one-step deletion protocol as described previously (*Longtine et al., 1998*). Pho92 and Gis2 were tagged with copies of the V5 epitope tag using tagging cassette described in *Chia et al., 2017*. *PAF1* was depleted using C-terminal auxin-inducible degron (AID) tag as described by *Nishimura et al., 2009*. For efficient depletion during meiosis, we used copper inducible *Oryza sativa* TIR (*osTIR*) ubiquitin ligase under the control of the *CUP1* promoter (*Chia et al., 2021*). The AID depletions of decay factors (*CAF40*, *RAT1*, *DCP2*, *RRP6*, *NOT5*, *NRD1*, *POP2*, *XRN1*) were described previously (*Dierks et al., 2021*). Seamless N-terminal tagging with superfolder GFP was performed as previously described (*Khmelinskii et al., 2011*). *URA3* gene served as a selection marker for positive selection in medium lacking uracil and counter selection in medium containing 5-fluoroorotic acid (5-FOA). Pho92-mNeonGreen fusions were constructed by tagging Pho92 with mNeonGreen tagging cassettes (*Argüello-Miranda et al., 2018*). Cells expressing Pho92-mNeonGreen fusions also harboured histone H2B fused with mCherry to determine nuclear localization. N-terminal 3 X Flag-tagged Pho92 constitutive expression in S288C was achieved using the plasmids described in *Zhang et al., 2017*. Cells expressing *CUP1-HA-PHO92*-WT or *CUP1-HA-PHO92-W177A* were generated after transformation of *pho92Δ* cells with p762 and p763 respectively, linearized with PmeI. Cells expressing *IME4^WT* or *IME4^CD* were generated after transformation with p782 and p783 (linearized with AgeI) in *ime4Δ* and genetic crosses. The strain genotypes are listed in *Supplementary file 1*.

### Growth conditions
All experiments were performed at 30 °C in a shaker incubator at 300 rpm. Cells were grown in YPD [1.0% (w/v) yeast extract, 2.0% (w/v) peptone, 2.0% (w/v) glucose, and supplemented with uracil (2.5 mg/l) and adenine (1.25 mg/l)].

For obtaining exponentially growing S288C cells for co-immunoprecipitations, yeast cells were grown in YPD to saturation overnight, diluted to $OD_{600}$ = 0.2 and subsequently were harvested after two to three doublings.

To induce entry into meiosis in SK1, a standard protocol for sporulation was followed. Cells were grown till saturation for 24 hr in YPD, then diluted at $OD_{600}$=0.4 to pre-sporulation medium BYTA [(1.0% (w/v) yeast extract, 2.0% (w/v) bacto tryptone, 1.0% (w/v) potassium acetate, 50 mM potassium phthalate] grown for about 16 hr, subsequently centrifuged, washed with sterile milliQ water, centrifuged again, re-suspended at $OD_{600}$=1.8 (with an exception for all CLIP experiments where cells were resuspended at $OD_{600}$=2.5) in sporulation medium (SPO) [0.3% (w/v) potassium acetate and 0.02% (w/v) raffinose)] and incubated at 30 °C. Cells were collected for protein, RNA, and DAPI staining analyses at the desired meiotic time points via pelleting at 1500 g, 2 min. To induce meiosis in S288C and A364A, cells were moved from saturated YPD to SPO medium and samples collected for DAPI counting every 24 hr until 72 hr.

To initiate sporulation synchronously from SK1 cells expressing *IME1* from *CUP1* promoter, 50 μM $CuSO_4$ was added to the medium 2 hr after shifting them to SPO. In cells expressing *NDT80* from *GAL1* promoter (*pGAL-NDT80*) and the transcription factor *GAL4*-ER, 1 μM β-estradiol was added at the same time $CuSO_4$.

For the m6A pull-down experiment, synchronous sporulation was induced following the method previously described (*Chia and van Werven, 2016*). Approximately 0.05 OD of exponentially growing yeast were inoculated into reduced glucose YPD. Cells were cultured overnight and washed in milli Q water before suspending them in SPO medium at $OD_{600}$ of 2.5. After 2 hr, 50 μM of $CuSO_4$ was added to induce expression from the *CUP1* promoter and initiate sporulation synchronously.

To enable efficient Paf1 depletion (*PAF1-AID*) for microscopy experiments, 1 mM of indole-3-acetic acid (IAA) and 50 μM of $CuSO_4$were added 2 hr after cells were shifted to SPO. For mRNA decay machinery depletion mutants, IAA and $CuSO_4$were added at 4 hr after shifting the cells to SPO and samples collected at 6 hr in SPO. As control, same volume of dimethyl sulphoxide (DMSO) was added to yeast cells.

## DAPI counting and spore viability

Cells were collected from sporulation cultures, pelleted via centrifugation (1500 g, 2 min), and fixed in 80% (v/v) ethanol for a minimum of 2 hr at 4 °C. The cells were pelleted again (1500 g, 2 min) and resuspended in PBS with 1 μg/ml DAPI. The proportion of cells containing one or multiple nuclei were counted using a fluorescent inverted microscope.

Tetrad/ dyad counts S288C and A364A: WT and *pho92Δ* strains were patched on YPD plates and incubated for 2 days. The patches were transferred on SPO plate and further incubated for 1 week at room temperature. Colonies were suspended in 10 μL of water and observed under 40 x objective (Axiostar Plus Zeiss). Number of packaged spores per ascus were counted for 200 asci.

Spore viability SK1: Diploid cells were patched from YPD agar plates to SPO agar plates and incubated for 3 days at 30 °C or 37 °C. Subsequently at least 40 tetrads (160 spores) were dissected and incubated at 30 °C for 72 hr on YPD agar plates before spore survival was assessed.

## RNA extraction

Yeast cells were harvested for RNA extraction by centrifugation, then washed once with sterile water prior to snap-freezing in liquid nitrogen. RNA was extracted from frozen yeast cell pellets using Acid Phenol:Chloroform pH 4.5 and Tris-EDTA-SDS (TES) buffer (0.01 M Tris-HCl pH 7.5, 0.01 M EDTA, 0.5% w/v SDS), by rapid agitation (1400 RPM, 65 °C for 45 min). After centrifugation, the aqueous phase was obtained and RNA was precipitated at –20 °C overnight in ethanol with 0.3 M sodium acetate. After centrifugation and washing with 80% (v/v) ethanol solution, dried RNA pellets were resuspended in DEPC-treated sterile water. Samples were further treated with rDNase (cat no 740.963, Macherey-Nagel) and column purified (cat no 740.948, Macherey-Nagel).

## RT-qPCR

For reverse transcription, ProtoScript II First Strand cDNA Synthesis Kit (New England BioLabs) was used and 500 ng of total RNA was provided as template in each reaction. qPCR reactions were prepared using Fast SYBR Green Master Mix (Thermo Fisher Scientific) and transcript levels were

quantified from the cDNA on Quantstudio 7 Flex Real Time PCR instrument. Signals were normalised over *ACT1*. Oligo sequences are listed in *Supplementary file 3*.

## RNA-Seq

For RNA sequencing, 1 μg of total yeast total RNA was used. Libraries were prepared using the KAPA RNA hyperPrep kit (KK8540, Roche) according to the manufacturer's instructions. Libraries were sequenced on an Illumina HiSeq 4000 to an equivalent of 75 bases single-end reads, at a depth of approximately 20 million reads per library.

Adapter trimming was performed with cutadapt (version 1.9.1) (*Martin, 2011*) with parameters `'--minimum-length=25 --quality-cutoff=20` a AGATCGGAAGAGC -A AGATCGGAAGAGC'. BWA (version 0.5.9-r16) (*Liao et al., 2014*) using default parameters was used to perform the read mapping independently to both the *S. cerevisiae* (assembly R64-1-1, release 90) genome. Genomic alignments were filtered to only include those that were primary, properly paired, uniquely mapped, not soft-clipped, maximum insert size of 2 kb and fewer than three mismatches using BamTools (version 2.4.0; *Barnett et al., 2011*). Read counts relative to protein-coding genes were obtained using the featureCounts tool from the Subread package (version 1.5.1) (*Liao et al., 2014*). The parameters used were '-O --minOverlap 1 --nonSplitOnly --primary -s 2 p -B -P -d 0 -D 1000 C --donotsort'.

Differential expression analysis was performed with the DESeq2 package (version 1.12.3) within the R programming environment (version 3.3.1) (*Love et al., 2014*). An adjusted p-value of ≤ 0.01 was used as the significance threshold for the identification of differentially expressed genes.

## Western blotting

Of yeast cells, 3.6 ODs were pelleted from cultures via centrifugation (1500 g, 2 min) for protein expression analysis. Proteins were precipitated from whole cells via trichloroacetic acid (TCA) incubation for a minimum of 2 hr at 4 °C. TCA was completely removed from the pellet via an acetone wash. Cells were lysed with glass beads using a bead beater and lysis buffer (50 mM Tris, 1 mM EDTA, and 2.75 mM DTT). Proteins were denatured in 3 X sample buffer (9% (w/v) SDS and 6% (v/v) β-mercaptoethanol) at 100 °C and separated by SDS/PAGE (4–12% gels). A PVDF membrane was used for protein transfer. Blocking was performed using 5% (w/v) dry skimmed milk. Proteins were detected using an ECL detection kit or using infra-red fluorescent antibodies visualised with LiCor CLx.

## Recombinant protein expression and purification

BL-21 DE3 were used for GST-fusion protein expression. Cells were grown in TB medium supplemented with 100 μg/mL ampicillin. IPTG was added to a final concentration of 0.5 mM, after which protein expression was induced for 4 hr at 30 °C. Cells were then harvested, and resuspended in lysis buffer (50 mM Tris-HCl, pH 8.0, 15% glycerol, 1 mM EDTA, 0.5 mM PMSF, 1 mM DTT, 100 mM NaCl, 1% Triton X-100, 0.25% NP-40, protease inhibitor cocktail, and lysozyme to a final concentration of 0.5 mg/ml). Cells were lysed by freeze-thaw cycles and short sonication. Bacterial debris was removed by ultra-centrifugation at 55,000 r.p.m. for 60 min at 4 °C, after which soluble extracts were aliquoted and snap-frozen in liquid nitrogen until further use.

## Dimethyl-labelling based m6A RNA pulldowns

RNA pull downs were performed using unlabelled lysates followed by dimethyl labelling essentially as described in *Edupuganti et al., 2017*. RNA probes used for this experiment are prepared in the lab of Thomas Carell (*Edupuganti et al., 2017*) and listed in *Supplementary file 3*. Forty μl of suspended beads (Dynabeads M280 Streptavidin) for two reactions (heavy and light, 20 μl per pull down) were first washed twice in 1 ml RNA binding buffer (50 mM HEPES-HCl, pH 7.5, 150 mM NaCl, 0.5% NP40 (v/v), 10 mM $MgCl_2$) and then incubated with RNase inhibitor RNasin plus (Promega) in RNA binding buffer (100 μl buffer with 0.8 units of RNasin/μl) for 15 min on ice. After removal of the buffer, beads were preblocked with yeast tRNA (100 μg/mL; AM7119; Life Technologies) in RNA binding buffer for 1 hr at 4 °C on a rotation wheel. The preblocked beads were washed twice with RNA binding buffer and then incubated with 5 μg of biotinylated RNA probe (per pulldown) diluted with RNA binding buffer to a final volume of 600 μl for 30 min at 4 °C in a rotation. The beads were washed once with 1 mL of RNA wash buffer (50 mM HEPES-HCl, pH 7.5, 250 mM NaCl, 0.5% NP-40 and 10 mM $MgCl_2$) and twice with protein incubation buffer (10 mM Tris-HCl, pH 7.5, 150 mM KCl, 1.5 mM $MgCl_2$, 0.1%

(v/v) NP-40, 0.5 mM DTT, and complete protease inhibitors (HALT)). During the last wash, beads in each vial, were split into two tubes one each for light and heavy labelling.

Beads containing immobilized RNA were then incubated with 500 ug of meiotic yeast extracts each in a total volume of 600 µl of protein binding buffer. The incubation reaction also contained 30 µg of yeast tRNA to prevent nonspecific binding, and RNasin. The reactions were incubated at room temperature for 30 min and then for 90 min on a rotation wheel at 4 °C. The beads were then washed three times with protein incubation buffer and twice with ice-cold PBS to remove detergent from the beads.

Proteins were on-bead digested with trypsin. Briefly, beads were resuspended in 100 µl of elution buffer (50 mM Tris, pH 8.5, 2 M urea and 10 mM DTT) and then incubated for 20 min at room temperature in a thermoshaker at 1100 r.p.m. Iodoacetamide was then added to a final concentration of 55 mM, and the mixture was incubated for 10 min in a thermoshaker (1100 r.p.m.) at room temperature in the dark. Proteins were then partially digested from the beads by the addition of 250 ng of trypsin for 2 hr at room temperature in a thermoshaker in the dark. After incubation, the supernatant was collected in a separate tube. The beads were then incubated with 50 µL of elution buffer for 5 min at room temperature in a thermoshaker (1100 r.p.m.). A total of 100 ng of fresh trypsin was added to the pooled eluates, and proteins were digested overnight at room temperature.

Tryptic peptides from individual pulldowns obtained after on-bead digestion were differentially labelled with dimethyl isotopes ($CH_2O$ or $CD_2O$) essentially as described (***Boersema et al., 2009***). Forward and reverse reactions were set up by mixing labelled peptides as follows: Forward reaction: control probe, $CH_2O$ (light); m6A probe, $CD_2O$ (medium) and Reverse reaction: control probe, $CD_2O$ (medium); m6A probe, $CH_2O$ (light). The reverse experiment represented a biological-replicate label swap. Labelled reactions were acidified to pH <2 with TFA (10%) and desalted with C18 Stage tips before MS analyses on the Orbitrap Fusion Tribrid mass spectrometer (Thermo), a similar LC gradient was used, and samples were measured in top-speed mode. Data processing was done with the MaxQuant software. Raw data were analyzed with DimethylLys0 and DimethylLys4 (light and medium) labels and matching between runs was enabled. UniProt database for *S. cerevisiae* downloaded on 13 July 2014 was used for identification. To identify significant interactors, normalized ratios from MaxQuant output tables for the forward and reverse pulldowns were plotted.

For validation of m6A readers identified by mass spectrometry, control or m6A oligos (Synthesized by https://biomers.net/ GmbH, listed in ***Supplementary file 3***) coupled to Streptavidin beads were incubated as above with either bacterial lysates expressing the respective recombinant proteins (GST- Gis2, Full length GST-Pho92 or truncation mutants of GST-Pho92) or SK1 yeast meiotic lysates expressing V5-tagged Gis2. After washes with protein incubation buffer, proteins bound to the beads were eluted by boiling for 5 min in laemmli buffer. The eluates were loaded onto SDS/ Polyacrylamide gels and assessed either by Coomassie staining or western blotting.

## m6A individual nucleotide resolution UV crosslinking and immunoprecipitation (miCLIP)

The miCLIP was adapted from ***Linder et al., 2015***, while using the iiCLIP library preparation protocol (***Lee et al., 2021***). Total RNA extracted as above was then used for polyA selection with oligodT dynabeads (cat no 61005, Life Technologies) according to manufacturer's instructions and up to 10 µg polyA RNA was fragmented to around 150–200 nt with fragmentation reagent (AM8740) in a 20 µl reaction and heat to 70 °C for 4 min. The reaction is then stopped by addition of stop solution provided in the kit. Four ug mRNA was added to 450 µl immunoprecipitation buffer IP (50 Mm Tris pH 7.4, 100 mM NaCl, 0.05% NP-40) and incubated with 5 µg anti-m6A (ab151230) for 2 hr at 4 °C with rotation. polyA RNA was cross-linked twice to the antibody with 2x0.15 $J/cm^2$ UV light (254 nm) in a Stratalinker and then incubated with 30 µl protein G beads for 1.5 hr at 4 °C with rotation. Bead bound antibody-RNA complexes recovered on a magnetic stand and washed twice with high-salt buffer (50 mM Tris pH 7.4, 1 M NaCl, 1 mM EDTA, 1% NP-40, 0.1% SDS, 0.5% deoxycholate), twice with immunoprecipitation buffer and twice with polynucleotide kinase PNK buffer (20 mM Tris, 10 mM $MgCl_2$, 0.2% Tween 20). Beads were resuspended in 20 µl of the following mixture: 4 µl 5 x PNK pH 6.5 buffer (350 mM Tris-HCl, pH 6.5, 50 mM MgCl2, 5 mM dithiothreitol), 0.5 µl PNK (NEB M0201L), 0.5 µl RNasin, 15 µl water and incubated for 20 min at 37 °C in thermomixer at 1100 rpm. Then washes with 1 ml PNK buffer, 1 ml high-salt wash, 1 ml PNK buffer followed and beads were resuspended in 100 µl

PNK buffer, moved to a new tube, place on magnet, removed supernatant and washed again once with 1 ml PNK buffer. Supernatant was removed and beads were resuspended in 25 µl of the ligation mix: 7.55 µl water, 3 µl 10 X ligation buffer [500 mM Tris-HCl, pH 7.5, 100 mM MgCl2, 0.8 µl 100% DMSO, 2.5 µl RNA ligase –high concentration (M0437 NEB)], 0.5 µl PNK (NEB M0201L), 0.4 µl RNasin (NEB), 1.25 µl pre-adenylated adaptor L3-App (AGATCGGAAG_1,AGCGGTTCAG_2, 20 µM), 9 µl 50% PEG8000 and incubated overnight at 16 °C in thermomixer at 1100 rpm. Washes followed with 2x1 ml high-salt wash and 1 ml PNK buffers. PNK buffer was removed and 20 µl of the removal mix [12.5 µl Nuclease-free H2O, 2 µl NEB Buffer 2, 0.5 ul Deµdenylase (NEB M0331S), 0.5 µl RecJ endonuclease (NEB M0264S), 0.5 µl RNasin, 4 µl PEG400] added. Incubation followed for 1 hr at 30 °C, then for 30 min at 37 °C whilst shaking at 1100 rpm. The beads were washed with 2x1 ml high-salt wash and 1 ml PNK buffer. Then the sample was split in two fractions. 10% was labelled with [gamma-P32] ATP ((γ-$^{32}$P)-ATP, SRP-301, Hartmann Analytic) in order to visualize mRNA-antibody complexes, the rest was not labelled and used for library preparation. 10% of the beads were collected, 4 µl of hot PNK mix [0.2 µl PNK (NEB M0201L), 0.5 µl 32P-γ-ATP, 0.4 µl 10 x PNK buffer (NEB), 2.9 µl water] was added after removal of supernatant and incubated for 5 min at 37 °C in thermomixer at 1100 rpm. Supernatant was removed and 100 µl PNK buffer was added, incubated in thermomixer at 37 °C /1100 rpm for 1 min, and again supernatant was removed.

SDS-PAGE and western blotting of labelled and non-labelled fractions followed with 4–12% NuPAGE Bis-Tris gel and MOPS buffer from Invitrogen and nitrocellulose membrane. The part of the membrane containing the signal was determined from the labelled fraction but the membrane was excised (75–150 kDa) from the non-labelled lane and cut into several pieces. Ten µl proteinase K (Roche, 03115828001) was added in 200 µl PK + SDS (10 mM Tris-HCl, pH 7.4, 100 mM NaCl, 1 mM EDTA, 0.2% SDS) buffer and to the nitrocellulose piece and incubation followed with shaking at 1100 rpm for 60 min at 50 °C. The solution was moved to another tube, 200 µl Phenol:Chloroform:Isoamyl Alcohol (Sigma P3803) added, mixed, and added to a pre-spun 2 ml Phase Lock Gel Heavy tube (713–2536, VWR). Incubation followed for 5 min at 30 °C with shaking at 1100 rpm. The phases were separated by centrifugation for 5 min at 13,000 rpm at room temperature. 800 µl chloroform was added to the top phase of the Phase Lock Gel Heavy tube and after centrifugation for 5 min at 13,000 rpm at 4 °C the aqueous layer was transferred into a new tube. The RNA was precipitated by addition of 0.75 µl glycoblue (Ambion, 9510), 20 µl 5 M sodium chloride and 500 µl 100% ethanol, followed by overnight incubation at –20 °C and spin for 20 min at 15,000 rpm at 4 °C. The pellet was washed with 0.9 ml 80% ethanol and resuspended in 5.5 µl H$_2$O.

1 µl primer irCLIP_ddRT_## 1 pmol/µl and 0.5 µ l dNTP mix (10 mM) were added to the resuspended 5.5 µl RNA pellet. Incubation at 65 °C for 5 min followed. Then the RT mix was added, 2 µl 5 x SSIV buffer (Invitrogen), 0.5 µl 0.1 M DTT, 0.25 µl RNasin, 0.25 µl Superscript IV (Invitrogen), and the RT reaction followed (25 °C 5 min, 50 °C 5 min, 55 °C 5 min). One µl of RnaseH (NEB, M0297S): RnaseA (Ambion, AM2274) mix was added and incubated at 37 °C for 15 min. Agencourt AMPure XP beads (3 x the volume of the RT reaction) added directly to the RT reaction and isopropanol to 1.7xvolume of the RT reaction. After 5 min incubation at RT, the beads were collected on magnetic stand, removed supernatant, 200 µl 85% ethanol added and incubated for 30 s, then removed the supernatant. The last step was repeated and the beads were left to dry on the magnetic stand and finally eluted with 8 µl water.

Circularization followed with addition of 2 µl of circligase II mix (1 µl 10 x CircLigase Buffer II, 0.5 µl CircLigase II, 0.5 µl 50 mM MnCl2 – kit from Epicentre) and the reaction incubated at 60 °C for 2 hr. Ampure bead clean up was repeated as above and the cDNA eluted in 10 µl water. The PCR reaction [1 µl cDNA, 0.25 µl primer mix P5Solexa/P3Solexa 10 µM each, 5 µl Phusion HF Master mix (M0531S), 3.75 µl water] was performed for 20 cycles with the following programme 98 °C 40 s, 20 x(98 °C 20 s/65 °C 30 s/72 °C 45 s), 72 °C 3 min, the products run on 6% TBE gel and stained with SYBR green I.

cDNA was excised in the range of 145–400 nt and the gel fragment was placed inside a 1.5 ml tube with holes in the bottom and placed in a 2 ml collection tube. After centrifugation at 13,000 rpm for 5 min, 500 µl of crush-soak gel buffer (500 mM NaCl, 1 mM EDTA, 0.05% SDS) was added to the gel and incubated at 65 °C for 2 hr with thermomixer settings of 15 s at 1000 rpm, 45 s rest. The liquid portion of the supernatant was transferred into a Costar SpinX column (Corning Incorporated, 8161) into which two 1 cm glass pre-filters (Whatman 1823010) were placed and centrifuged at 13,000 rpm

for 1 min. The solution was collected and 1 μl glycoblue, 50 μl 3 M sodium acetate, pH 5.5 and 1.5 ml 100% ethanol were added. The RNA was precipitated overnight at –20 °C followed by centrifugation for 30 min at 15,000 rpm at 4 °C. The cDNA pellet was washed once with 0.9 ml 80% EtOH, resuspended in 10–15 μl RNase free $H_2O$ and submitted for sequencing.

The entire protocol was also performed with 1 μg input mRNA omitting IP, P32 labelling, SDS-PAGE/western blotting steps. We moved from adapter removal to reverse transcription. An RNA clean up step using RNA clean and concentrator kit (Zymo Research R1016) was included each time after adapter ligation and adapter removal.

## UV-C (254 nm) and 4-ThioUracil-UV-A (365 nm) individual nucleotide resolution UV Crosslinking and Immunoprecipitation (iCLIP)

We used the improved iCLIP (iiCLIP) protocol (*Lee et al., 2021*). 250 ml of SK1 cells were cultured as stated above and grown for 4 h in SPO. For UV-A 4-TU-iCLIP, SK1 cells expressing the *FUR4* transgene (to facilitate uptake of 4-thiouracil) were grown for 4 h in SPO containing 0.5 mM 4-thio uracil. After 4 hours, cells were collected in 10- or 20 ml PBS in a petri dish on ice and crosslinked with 0.8 J/cm² UV light (254 nm) or 12 J/cm² UV light (365 nm) in a Stratalinker 2400. Crosslinked cells were collected and resuspended in 2 ml lysis buffer (50 mM Tris-HCl pH 7.5, 100 mM NaCl, 0.5% Na-DOC, 0.1% SDS, 0.5% NP-40, 1 mM DTT, 1 X Protease inhibitor cocktail (HALT)) and pebbles prepared in liquid nitrogen. Cells were subjected to cryogenic lysis by freezer mill grinding under liquid nitrogen (SPEX 6875D Freezer/Mill, standard program: 15 cps for 6 cycles of 2 min grinding and 2 min cooling each). The resultant yeast 'grindate' powder was resuspended in equal volume of lysis buffer and left to tumble in the cold for an hour. The cell lysate was initially clarified by a short 5-min centrifugation at 1500 rpm at 4 °C and then further purified by ultra-centrifugation at 45,000 rpm for 1 hr at 4 °C. The clarified lysates were subjected to DNase (2 μl of Turbo DNase per mg protein) and RNase (0.05 units/ mg lysate) treatment for 3 min at 37 °C and then transferred to ice. Dynabeads (Protein G) were washed twice in lysis buffer before incubating 10 μg of V5 antibody (for each IP) with the beads for 1 hr at room temperature. Subsequently, the beads were washed thrice in lysis before adding the lysates to the beads-antibody mix. The immunoprecipitation was carried out overnight at 4 °C and from here onwards the same steps as in the miCLIP methods given above were followed starting with the wash with high-salt buffer and including the library preparation with the only exception that RNA-protein complexes in this case were excised out of the membrane from above the size of the respective protein bands.

## iCLIP/miCLIP pre-processing

Reads were demultiplexed using iCount demultiplex, and subsequently trimmed for adapter sequences and also for PHRED score >30 using Trim Galore! (*Krueger, 2017*). Due to the high ncRNA content in miCLIP libraries a sequential mapping strategy was used for all libraries, which is available as a Snakemake (*Köster and Rahmann, 2012*) pipeline from https://www.github.com/ulelab/ncawareclip/, (*Varier, 2022* copy archived at swh:1:rev:11b92a6302c5738f1c8c6da1f8b4aeb405082375). Mapping was first to representative *Saccharomyces cerevisiae* snRNA and rRNA sequences downloaded from NCBI (*Pruitt et al., 2005*), followed by mature tRNA sequences (3' CCA and 5' G added) and the SacCer3 mitochondrial chromosome before being mapped to the SK1 MvO genome (available from http://cbio.mskcc.org/public/SK1_MvO/). PCR duplicates were removed using the unique molecular identifiers (UMIs). The start positions of uniquely mapping reads were taken as crosslinks. Detailed annotations were taken from *Chia et al., 2021*. tRNA annotations were downloaded from UCSC table browser, which sources the annotations from GtRNAdb (*Karolchik et al., 2004*; *Chan and Lowe, 2016*). SK1 annotations were supplemented with 3' and 5' UTR annotations derived (*Chia et al., 2021*). All genome browser screenshots display stranded crosslinks per million (CPM) normalised bigWig files. Crosslinks at each position were divided by the total number of genomic crosslinks in the sample multiplied by one million.

## iCLIP/miCLIP differential analysis

Peaks were called on iCLIP and miCLIP data with Clippy v1.2.0 (https://github.com/ulelab/clippy/releases/tag/v1.2.0; *Ulelab et al., 2021*; copy archived at swh:1:rev:e89d1d953611a0baebd56f-9c178eebecd782799f), specifically with settings -n 20 -up 50 -down 100 x 2.75 -hc 0.8 mg 10 mb

10 no_exon_info. Reproducible, high-quality replicate samples, as determined by principal component analysis (PCA), replicate correlations, library size and regional crosslink locations, were combined for peak calling and then individual sample coverage over these peaks was calculated using Bedtools map (*Quinlan, 2014*). Peaks were filtered to have at least 5 cDNAs in 3 WT or 3 *ime4-Δ* replicates, to come to a preliminary list of binding sites.

To determine Ime4p-dependence, WT vs. *ime4Δ* samples for Pho92 iCLIP, Gis2 iCLIP and m6A miCLIP were compared using DeSeq2, whilst controlling for gene expression changes by including measurements from mock miCLIP samples as a contrast in the linear model (*Love et al., 2014*; *McIntyre et al., 2020*). Genes with less than 20 cDNA counts across three replicates were discarded from the analysis. p Values were calculated using a likelihood-ratio test. A stringent threshold of log2FoldChange <= −2 and adjusted p value of<0.001 was used to determine differentially bound sites for Pho92 and Gis2 iCLIP. For m6A miCLIP a threshold of log2FoldChange <= −1 and adjusted p value of<0.001 were used. Due to the high depth of the iCLIP datasets, sites were further filtered based on a DeSeq2 base mean >200, which is a measure of coverage in both iCLIP and mock iCLIP samples. The value is calculated as the average of the normalized cDNA counts per peak from all samples, divided by their size factors. Peak assignment to transcriptomic regions was performed using the following hierarchy to resolve any overlapping annotations: snoRNA >ncRNA > STOP codon >3' UTR >5'UTR >last 100nt CDS >first 100nt CDS >CDS > intergenic.

## Published m6A sites definition
To come to a group of published m6A sites to compare our data against, all Mazter-Seq sites with confidence group >1 and all m6A-Seq sites were taken (*Schwartz et al., 2013*; *Garcia-Campos et al., 2019*). To robustly map these to the MvO SK1 genome assembly, all intervals were expanded to 150nt, sequences retrieved and BLAST used to get mappings. Mappings were filtered to those that were unique and in the case of Mazter-Seq, perfectly aligned to an 'ACA' sequence. To create a consensus list, any m6A-seq region that overlapped with Mazter-Seq site(s) was removed, under the assumption that the signal was representing the sites detected at higher resolution by Mazter-Seq - although it is possible that adjacent non-ACA sequence context m6A sites would not be represented in the list. This procedure resulted in a list of 1297 m6A regions.

## Distance to nearest m6A
Distances between peak sets were calculated using bedtools closest, with parameters -s -t first -d (*Quinlan, 2014*).

## Go term enrichment
Gene list enrichment analysis was performed using YeastEnrichr, specifically using KEGG 2019 pathways (https://maayanlab.cloud/YeastEnrichr/) (*Chen et al., 2013*; *Kuleshov et al., 2016*).

## Motif analysis
Motifs were discovered in peak regions by resizing all peaks to 100nt and obtaining their fasta sequences to submit to STREME (*Bailey, 2021*). Either shuffled sequences or another peak set were used as background, as indicated in the main text. Motifs were plotted around the centre of peaks.

## Metaprofiles
Bigwigs were generated from bedgraphs using UCSC BedGraphToBigwig (*Kuhn et al., 2013*) and metaprofiles were generated around regions of interest using DeepTools ComputeMatrix and PlotHeatmap (*Ramírez et al., 2014*). Further integration and plotting was performed using R with ggplot2, dplyr, data.table, stringr and cowplot packages (*Wickham, 2010*; *Wickham et al., 2015*; *Wickham, 2016*; *Dowle et al., 2019*; *Wilke and Wickham, 2019*).

## Determining Pho92 intronic overlaps
Due to the sparsity of SK1 genome annotations, many known introns that are present in SacCer3 annotations are not present in SK1 annotation. Therefore, to determine intronic overlaps, all Pho92-bound genes with introns in SacCer3 annotations were manually inspected alongside miCLIP-input

and checked for whether Pho92 peaks overlapped regions suspected to be introns due to: (a) sharp drop in miCLIP-input signal in the region, (b) corresponding intron in SacCer3 annotations.

## Immunoprecipitation and mass spectrometry

Extracts were prepared in a mild buffer (50 mM Tris-HCl pH 7.5, 100 mM NaCl, 10 mM MgCl2, 0.1% Na-DOC, 0.1% SDS, 0.1% NP-40, 1 mM DTT, 1 X Protease inhibitor cocktail (HALT)) using cryogenic lysis as detailed above from S288C expressing *pCYC1*- FLAG-Pho92, *pPYK1*- FLAG-Pho92 and SK1 expressing Pho92-V5. Untagged S288C and SK1 strains were used as controls. Extracts were incubated with either Flag (M2-agarose) or V5 agarose for 2 hr at 4 °C. Washes were done with the same buffer as above but with an addition of 0.2% NP-40 instead. Elutions from Flag beads were done with 2 mg/ml flag -peptide. All eluates were combined and samples were further processed using the PreOmics iST (inStageTip) protocol and the manufacturer's protocol was followed. Eluted peptides were dried by vacuum centrifugation. Samples were solubilised in 20 µl of 0.1% TFA and injected in technical triplicate Orbitrap Fusion Lumos Tribrid mass spectrometer.

For V5 pull downs, after the V5 peptide elutions (which were found to be inefficient), a further laemmli buffer elution from the beads by boiling was performed. Laemmli buffer eluate samples were processed using the single-pot, solid phase-enhanced, sample preparation (SP3) technology. Paramagnetic beads were added to the tryptic digests and peptide eluates were purified using C18 stagetips. Samples were injected in technical triplicates on the QExactive instrument.

All raw files were analysed in MaxQuant v1.6.0.13 against the 2019 SwissProt *S. cerevisiae* protein database & additional data analyses were performed in Perseus v1.4.0.2.

## Co-Immunoprecipitation and GST pull downs

Cells were resuspended in lysis buffer (50 mM Tris-HCl pH 7.5, 150 mM NaCl, 10 mM MgCl$_2$, 0.5% Na-DOC, 0.1% SDS, 0.2% NP-40, 1 mM DTT, and 1 X HALT protease inhibitor) and lysed using Zirconia beads in beadbeater (BioSpec). The lysates were cleared via centrifugation (16,000 g at 4 °C for 20 min). Flag-Pho92 and associated proteins were pulled down using anti-Flag M2-conjugated agarose beads by incubating at 4 °C for 2 hr. Unbound proteins were washed from the beads with wash buffer (50 mM Tris-HCl pH 7.5, 1 M NaCl, 10 mM MgCl$_2$, 0.5% Na-DOC, 0.1% SDS and 0.5% NP-40). Bound proteins were eluted by competition with an excess of free Flag peptide. Samples were analysed by western blotting.

Soluble GST-tagged fusion proteins expressed and prepared as above were immobilized on glutathione-agarose beads and subsequently used for binding proteins from yeast extracts expressing HA-tagged Paf1C or Rap1 prepared as above for co-immunoprecipitations. The bound proteins were eluted off the GA beads and analyzed by Western blotting by probing with anti- HA antibody.

## Global whole-cell proteome profiling

Cells were collected by centrifugation, re-suspended in cold 5% v/v TCA and incubated on ice for 30 min. Samples were washed with acetone, then completely air-dried. The dried pellet was resuspended in protein breakage buffer (50 mM Tris (pH 7.5), 1 mM EDTA, 2.75 mM dithiothreitol (DTT), HALT protease inhibitors) and disrupted using 0.5 mm glass beads for 2 min in a Mini Beadbeater (Biospec). To the lysate, 3 X concentrated modified laemmli buffer (187.5 mM Tris (pH 6.8), 6.0% v/v β-mercaptoethanol, 30% v/v glycerol, 9.0% v/v SDS) was added along with protease inhibitors (HALT) to result in a final concentration of 62.5 mM Tris, 2% β-mercaptoethanol, 10% glycerol and 3% SDS and denatured at 95 °C for 5 min.

Reduction and alkylation by the addition of 20 mM TCEP and 40 mM chloroacetamide to the provided protein lysates was carried out for 10 min at 70 °C. Protein amounts were confirmed, following an SDS–PAGE gel of 4% of each sample against an in-house cell lysate of known quantity. A volume corresponding to 50 µg of each sample was taken along for digestion. Proteins were precipitated overnight at −20 °C after addition of a 4×volume of ice-cold acetone. The following day, the samples were centrifuged at 20,800 x g for 30 min at 4 °C and the supernatant carefully removed. Pellets were washed twice with 1 ml ice-cold 80% (v/v) acetone in water then centrifuged at 20,800 x g at 4 °C. They were then allowed to air-dry before addition of 120 µl of digestion buffer (1 M Guanidine Hydrochloride, 100 mM HEPES, pH8). Samples were resuspended with sonication, LysC (Wako) was added at 1:100 (w/w) enzyme:protein, and digestion proceeded for 4 hr at 37 °C

with shaking (Eppendorf ThermoMixerC, thermoblock for 1.5 ml tubes, at 1000 rpm for 1 hr, then 650 rpm). Samples were then diluted 1:1 with Milli-Q water, and trypsin (Pierce) added at the same enzyme to protein ratio. Samples were further digested overnight at 37 °C with shaking (650 rpm). The following day, digests were acidified by the addition of TFA to a final concentration of 2% (v/v) and then desalted with Waters Oasis HLB μElution Plate 30 μm (Waters Corporation, Milford, MA, USA) in the presence of a slow vacuum. In this process, the columns were conditioned with 3×100 μl solvent B (80% (v/v) acetonitrile; 0.05% (v/v) formic acid) and equilibrated with 3×100 μl solvent A (0.05% (v/v) formic acid in Milli-Q water). The samples were loaded, washed 3 times with 100 μl solvent A, and then eluted into with 50 μl solvent B. The eluates were dried down with the speed vacuum centrifuge and dissolved at a concentration of 1 μg/μl in reconstitution buffer (5% (v/v) acetonitrile, 0.1% (v/v) formic acid in Milli-Q water).

Digested peptides were separated using the Dionex U3000 nano UHPLC system (Thermo) fitted with a trapping (PepMap Acclaim C18, 5 μm, 0.2 mm x 20 mm) and an analytical column (EasySpray PepMap C18, 2 μm, 75 μm x 500 mm), both held at 40 °C. The outlet of the analytical column was coupled directly to an Orbitrap Fusion Lumos Tribrid mass spectrometer (Thermo Fisher Scientific) using the EasySpray source. Peptides were eluted via a non-linear gradient from 100% aqueous/0.1% Formic acid to 40% Acetonitrile/0.1% Formic Acid in 98 min. Total runtime was 120 min, including clean-up and column re-equilibration. The RF lens was set to 30%. The spray voltage was set to 2.2 kV and source temperature 275 °C. Default charge state was set to 4+. For data acquisition and processing Tune version 3.3 was employed. Data Independent Acquisition (DIA) data was acquired with full scan MS spectra over the mass range 350–1650 m/z in profile mode in the Orbitrap with resolution of 120,000 FWHM. The filling time was set at maximum of 20ms with limitation of 1E6 ions. DIA MS2 scans were acquired with 34 mass window segments of differing widths across the MS1 mass range with a cycle time of 3 s. HCD fragmentation (30% collision energy) was applied and MS/MS spectra were acquired in the Orbitrap at a resolution of 30,000 FWHM over the mass range 200–2000 m/z after accumulation of 1E6 ions or after filling time of 70ms whichever occurred first. Data were acquired in profile mode.

DIA data were searched directly against the SwissProt *Saccharomyces cerevisiae* database (6721 entries) and a list of common contaminants using Direct DIA in Spectronaut (version 14, Biognosys AG, Schlieren, Switzerland). The following modifications were included in the search: Carbamidomethyl (C) (Fixed) and Oxidation (M)/Acetyl (Protein N-term; Variable). A maximum of 2 missed cleavages for trypsin were allowed. The identifications were filtered to satisfy FDR of 1% on peptide and protein level. The DirectDIA analysis resulted in 33,575 precursors and 3292 Protein Groups as full profiles across the six samples. Precursor matching, protein inference, and quantification were performed in Spectronaut using default settings. The candidate table was exported for further data visualisation.

## Chromatin immunoprecipitation

Chromatin immunoprecipitation (ChIP) experiments were performed as described previously (*Tam and van Werven, 2020*). Cells were fixed in 1.0% v/v formaldehyde for 20 min at room temperature and quenched with 100 mM glycine. Cells were lysed in FA lysis buffer (50 mM HEPES–KOH, pH 7.5, 150 mM NaCl, 1 mM EDTA, 1% Triton X-100, 0.1% Na-deoxycholate, 0.1% SDS and protease cocktail inhibitor (complete mini EDTA-free, Roche)) using beadbeater (BioSpec) and chromatin was sheared by sonication using a Bioruptor (Diagenode, 9 cycles of 30 s on/off). Extracts were incubated for 2 hr at room temperature with anti-V5 agarose beads (Sigma), washed twice with FA lysis buffer, twice with wash buffer 1 (FA lysis buffer containing 0.5 M NaCl), and twice with wash buffer 2 (10 mM Tris–HCl, pH 8.0, 0.25 M LiCl, 1 mM EDTA, 0.5% NP-40, 0.5% Na-deoxycholate). Subsequently, reverse cross-linking was done in 1% SDS-TE buffer + Ribonuclease A (10 ng/μl) (100 mM Tris pH 8.0, 10 mM EDTA, 1.0% v/v SDS) at 65 °C overnight. After 2 h of proteinase K treatment, samples were purified, and DNA fragments were quantified by real-time PCR using PowerUP SYBR green master mix (Thermo Fisher Scientific) using primers described in *Supplementary file 3*.

## Fluorescence microscopy and quantification

Cells were collected from sporulation cultures at desired time points, pelleted via centrifugation (1500 g, 2 min) and resuspended in approximately 100 μl of SPO media. Live cell image acquisition was conducted using a Nikon Eclipse Ti inverted microscope. Exposure times were set as follows:

500ms brightfield, 50ms GFP, and 50ms mCherry. An ORCA-FLASH 4.0 camera (Hamamatsu) and NIS-Elements AR software (Nikon) were used to collect images. Quantification of fluorescence signals was performed using FiJI software (*Schindelin et al., 2012*). ROIs were manually drawn around the periphery of each cell. The mean intensity in each channel per cell was multiplied by the cell area to obtain mean signal. The signal for each channel was corrected for cell-free background fluorescence in a similar way. Values for nuclear protein localisation were derived via the division of nuclear / whole cell signal. Whole cell value ranges were set for each time point between different strains for a fair comparison of nuclear signals. For the analyses, 150 cells were quantified per sample.

## Ribosome association by sucrose cushion

Ribosomal fractions were separated from soluble components by centrifugation through sucrose cushions (*Trotter et al., 2008*). Cell extracts were prepared from S288C 100 ml cells $OD_{600}$ 0.5 in YPD, collected and centrifuged after addition of 10 µg/ml cycloheximide. Cell pellets were washed with 1 ml CSB buffer [300 mM sorbitol, 20 mM Hepes (pH 7.5), 1 mM EGTA, 5 mM $MgCl_2$, 10 mM KCl, 10% (v/v) glycerol and 10 µg/ml cycloheximide, protease inhibitors] and resuspended in 500 µl CSB buffer, glass beads added and breakage followed in a bead beater for 3 min. Additional 300 µl CSB buffer were added and cell suspension was centrifuged for 4 min at full speed and 4 °C. The supernatant was moved to a fresh tube and centrifuged at 10,000 rpm for 15 min at 4 °C. 500 µl of supernatant was layered onto 400 µl 60% sucrose in CSB without sorbitol and centrifuged at 5500 rpm for 3.5 hr in TLA110 at 4 °C. The supernatant was precipitated with 20% TCA and this as well as the ribosomal pellet were resuspended each in 50 µl Laemmli sample buffer.

## Ribosome fractionation on sucrose gradients

Performed as previously described (*Ashe et al., 2000*). SK1 Cells were grown till saturation for 24 hr in YPD, then diluted at $OD_{600}$=0.4 to pre-sporulation medium BYTA grown for about 16 hr, subsequently centrifuged, washed with sterile Milli-Q water, centrifuged again, re-suspended at $OD_{600}$=1.8 in sporulation medium (SPO) and incubated at 30 °C. Yeast cells were harvested after 4 hr in SPO. S288C cells were collected at $OD_{600}$ 0.5 in YPD as for sucrose cushions. A total of 200 ml cells were collected with addition of cycloheximide to 100 µg/ml final concentration, washed and resuspended in 2 ml lysis buffer (20 mM Hepes pH 7.4, 2 mM MgAcetate, 100 mM KAcetate, 100 µg/ml cycloheximide, 500 µM DTT, PIC, RNase inhibitor), and pebbles prepared in liquid nitrogen. Cells were subjected to cryogenic lysis by freezer mill grinding under liquid nitrogen (SPEX 6875D Freezer/Mill, standard program: 15 cps for 6 cycles of 2 min grinding and 2 min cooling each). Yeast 'grindate' powder was stored at –80 °C. 0.5 ml yeast grindate powder was resuspended in 0.5 ml lysis buffer and centrifuged at 10,000 rpm for 5 min at 4 °C. Supernatant moved to a fresh tube and centrifuged at 10,000 rpm for 15 min at 4 °C. 300 µg of $A_{260}$ was loaded onto a 10–45% sucrose gradient. The sucrose solutions were in the following buffer: 10 mM Tris Acetate pH 7.4, 70 mM ammonium acetate, 4 mM MgAcetate. Polysome profiles were obtained from a Biocomp fractionator and samples collected for further analysis. For protein analysis 20% TCA was added to the fractions collected, protein precipitated and run on western blotting. For RNA-seq, polysome fractions were precipitated overnight at –20 °C with 400 µl isopropanol. The RNA pellet was precipitated and resuspended in 50 µl RNase-free water, 800 µl TES and 800 µl Acid phenol (see RNA extraction-miclip section) added and incubated for 5 min at RT. Further the RNA pellet was precipitated, DNase treated, and column purified as in RNA extraction/miclip section.

## Next generation sequencing

The NEBNext Ultra II Directional RNA Library Prep Kit for Illumina with the NEBNext Poly(A) mRNA Magnetic Isolation Module was used. Approximately 12 ng of RNA was used as input, according to manufacturer's instructions. Sequencing was carried out on the Illumina Hiseq 4000 or NovaSeq 6000 with single ended 100 bp reads.

## m6A-mRNA quantification by LC-MS/MS and ELISA

RNA extracted and DNase treated as described above and polyA RNA isolated after purification twice with oligodT dynabeads (Ambion, 61005) according to manufacturer's instructions. 375 ng polyA selected RNA in 20.5 µl $H2O$ was digested with addition of 2.5 µl of 10 X Buffer (25 mM ZnCl2,

250 mM NaCl, 100 mM NaAcetate), 2 µl of Nuclease P1 (1 U/µl) (Sigma, N8630) and incubated for 4 hr at 37 °C. Then 3 µl of 1 M ammonium bicarbonate, 1 µl Alkaline Phosphatase (NEB, M0525) and 1 µl of H2O were added. Incubation followed for 2 hr at 37 °C. 19 µl H2O and 1 µl of 5% Formic Acid added to the reaction mixture and filtered with a 1.5 ml microfuge tube containing a 0.22 µM inside filter. The sample was then injected (20 µl) and analysed by LC-MS/MS using a reverse phase liquid chromatography C18 column and a triple quadrupole mass analyser (Agilent 6470 or Thermo Scientific TSQ Quantiva) instrument in positive electrospray ionisation mode. Flow rate was at 0.2 ml/min and column temperature 25 °C with the following gradient: 2 min 98% eluent A (0.1% formic acid and 10 mM ammonium formate in water) and 2% eluent B (0.1% formic acid and 10 mM ammonium formate in MeOH), 75% A and 25% B up to 10 min, 20% A and 80% B up to 15 min, 98% A and 2% B up to 22.5 min. Nucleosides were quantified using the mass transitions of 281–150 for m⁶A and 268–136 for adenosine and a calibration curve of pure nucleosides standards.

For ELISA, we either used the EpiQuik m6A RNA methylation quantification kit from EpiGentek (P-9005) according to manufacturer's protocol or the protocol we developed (*Ensinck et al., 2022*). In our method 90 µl binding solution (ab156917) and 50 ng of twice polyA selected RNA (similar to LC-MS/MS above) were added to each well of a 96-well plate (ab210903). Three technical replicates were used for each sample and standard. The plate incubated at 37 °C for 2 hr. After incubation, binding solution and mRNA removed and wells washed four times with 200 µl PBSTween (0.1%) in each wash. A total of 100 µl of primary antibody (mRabbit A19841, 1:10000 in PBSTween) containing 0.5 µg/ml total RNA from *ime4Δ* mutant added and the plate incubated at RT for 1 hr. Following incubation, solution was removed and wells washed four times with 200 µl PBSTween (0.1%) each time, while incubating the plate with the third wash for 5 min at RT. 100 µl of secondary antibody (anti-rabbit 1:5000 in PBST, ab205718) added and incubation at RT for 30 min followed. Solution was removed and wells washed 5 times with 200 µl PBSTween (0.1%) each time, while incubating the plate with the third wash for 5 min at RT. 100 µl of developing solution (ab156917) added to the wells and left for 20–30 min before addition of 100 µl of stop solution (ab171524). Absorbance was recorded at 450 nm. Quantity of m6A was calculated using a calibration curve with standards of in vitro transcribed m6A-RNA and adenosine-RNA in the range of 0.0005–0.0125 ng for m6A-RNA with addition of 50 ng adenosine-RNA in each standard. All reagents bought from Abcam. I n vitro transcribed adenosine-RNA and m6A-RNA were synthesized with MEGAscript T7 Transcription kit (AM1333, Ambion) from ATP and N⁶-Methyl-ATP (NU1101L, 2B Scientific), respectively.

## m6A-mRNA stability assay

SK1 cells were grown till saturation for 24 hr in YPD, then diluted at OD600=0.4 to pre-sporulation medium BYTA and grown for about 16 hr, subsequently centrifuged, washed with sterile Milli-Q water, centrifuged again, re-suspended at OD600=1.8 in SPO. After 4 hr in SPO cultures were treated either with 3 µg/ml thiolutin (T2834, Cambridge Bioscience) or with 3 µg/ml thiolutin and 100 µg/ml cycloheximide (C4859, Merck) and samples collected at indicated timepoints. RNA extraction, polyA selection and m6A quantification either with ELISA or LC-MS/MS followed as described in relevant sections.

For mRNA stability of m6A containing targets that bind to Pho92, RT-qPCR was performed. Cells grown as above and after 4 hr in SPO were shifted to YPD as transcription of genes that are regulated by *IME1* is instantly shut down (*Chia et al., 2017*). Samples were collected at indicated time points and RT-qPCR performed as described above.

## Statistical analyses

Data statistics and statistical analyses indicated in the figure legends were computed using GraphPad Prism version 8.2.0 for Windows, GraphPad Software, San Diego, CA, USA, https://www.graphpad.com/. Data from the imaging experiments were analysed using paired parametric two-tailed Welch's t test with 95% confidence. Boxplots highlight median and quartiles 1 and 3. p-Values are indicated in the figures, where ns stands for no significant difference, *p<0.05, **p<0.01, ***p<0.001, and ****p<0.0001.

## Data and materials availability

Strains and plasmids are available upon request. The miCLIP, iCLIP and RNA-seq RAW and processed data are available at GEO accession GSE193561. Primer sequences used in the study are deposited

in *Supplementary file 3*. Processed data from the MS experiment in *Figure 1C* and *Figure 1—figure supplement 1B* are deposited in *Supplementary file 4*. Processed data from the miCLIP and iCLIP data are deposited in *Supplementary file 5*. Processed data from the IP-MS experiment in *Figure 4A* and *Figure 4—figure supplement 1C* are deposited in *Supplementary file 6*. Scripts used for analysing the iCLIP data can be found at https://www.github.com/ulelab/ncawareclip/, (*Varier, 2022* copy archived at swh:1:rev:11b92a6302c5738f1c8c6da1f8b4aeb405082375).

## Acknowledgements

We are grateful to the members of the laboratory of Folkert van Werven for fruitful discussions and critical reading of the manuscript. We thank members of Ule and Luscombe labs, specifically Julian Zagalak, Christoph Sadee, Patrick Toolan-Kerr, Igor Ruiz De Los Mozos, Paulo Gameiro, Federica Capraro, Andrew Steele and Flora Lee for sharing ideas, protocols and reagents towards the iCLIP and miCLIP experiments and analysis. We acknowledge Leon Chan, Elçin Ünal, and Martin Pool for sharing reagents and Schraga Schwartz and David Dierks for providing the m6A-seq2 data. We thank the Crick Advanced Sequencing, Proteomics, Metabolomics, Fermentation and Genomics Equipment Park Facilities for experimental support, specifically Phil East for helping with submission to GEO, James MacRae, Christoph Messner and Svend Kjaer for technical support. We also are grateful to the three reviewers for their comments and suggestions.The Vermeulen lab is part of the Oncode institute, which is partly funded by the Dutch Cancer Society (KWF). This research was funded in whole, or in part, by the Wellcome Trust (FC001203, FC010110, FC001134). For the purpose of Open Access, the author has applied a CC BY public copyright licence to any Author Accepted Manuscript version arising from this submission. This work was supported by the Francis Crick Institute (FC001203, FC010110, FC001134), which receives its core funding from Cancer Research UK (FC001203, FC010110, FC001134), the UK Medical Research Council (FC001203, FC010110, FC001134), and the Wellcome Trust (FC001203, FC010110, FC001134).Wellcome Trust (FC001203) – Radhika Varier, Theodora Sideri, Zornitsa Manova, Imke Ensinck, Alice Rossi, Folkert van Werven; CRUK (FC001203) - Radhika Varier, Theodora Sideri, Zornitsa Manova, Imke Ensinck, Alice Rossi, Folkert van Werven; Medical research council (FC001203) - Radhika Varier, Theodora Sideri, Zornitsa Manova, Imke Ensinck, Alice Rossi, Folkert van Werven; Wellcome Trust (FC010110) – Charlotte Capitanchik and Nicholas Luscombe; CRUK (FC010110) - Charlotte Capitanchik and Nicholas Luscombe; Medical research council (FC010110) - Charlotte Capitanchik and Nicholas Luscombe; Wellcome Trust (FC001134) – Enrica Calvani and Markus Ralser; CRUK (FC001134) - Enrica Calvani and Markus Ralser; Medical research council (FC001134) - Enrica Calvani and Markus Ralser; Dutch Cancer Society (KWF) - Raghu Edupuganti and Michiel Vermeulen.The funders had no role in study design, data collection and interpretation, or the decision to submit the work for publication. We also are grateful to the three reviewers for their comments and suggestions.

## Additional information

### Funding

| Funder | Grant reference number | Author |
| --- | --- | --- |
| Wellcome Trust | FC001203 | Radhika A Varier<br>Theodora Sideri<br>Zornitsa Manova<br>Alice Rossi<br>Imke Ensinck<br>Folkert J van Werven |
| Cancer Research UK | FC001203 | Radhika A Varier<br>Theodora Sideri<br>Zornitsa Manova<br>Alice Rossi<br>Imke Ensinck<br>Folkert J van Werven |

| Funder | Grant reference number | Author |
|---|---|---|
| Medical Research Council | FC001203 | Radhika A Varier<br>Theodora Sideri<br>Zornitsa Manova<br>Alice Rossi<br>Imke Ensinck<br>Folkert J van Werven |
| Wellcome Trust | FC010110 | Charlotte Capitanchik<br>Nicholas M Luscombe |
| Cancer Research UK | FC010110 | Charlotte Capitanchik<br>Nicholas M Luscombe |
| Medical Research Council | FC010110 | Charlotte Capitanchik<br>Nicholas M Luscombe |
| Medical Research Council | FC001134 | Enrica Calvani<br>Markus Ralser |
| Cancer Research UK | FC001134 | Enrica Calvani<br>Markus Ralser |
| Dutch Cancer Society | | Michiel Vermeulen<br>Raghu R Edupuganti |

The funders had no role in study design, data collection and interpretation, or the decision to submit the work for publication. For the purpose of Open Access, the authors have applied a CC BY public copyright license to any Author Accepted Manuscript version arising from this submission.

## Author contributions

Radhika A Varier, Conceptualization, Data curation, Formal analysis, Supervision, Methodology, Writing – original draft, Writing – review and editing; Theodora Sideri, Data curation, Software, Formal analysis, Validation, Methodology, Writing – original draft, Writing – review and editing; Charlotte Capitanchik, Data curation, Software, Formal analysis, Investigation, Methodology, Writing – original draft, Writing – review and editing; Zornitsa Manova, Alice Rossi, Data curation, Formal analysis; Enrica Calvani, Raghu R Edupuganti, Imke Ensinck, Peter Faull, Formal analysis, Methodology; Vincent WC Chan, Joanna Kirkpatrick, Formal analysis; Harshil Patel, Software, Formal analysis; Ambrosius P Snijders, Formal analysis, Supervision; Michiel Vermeulen, Markus Ralser, Supervision, Methodology; Jernej Ule, Resources, Supervision, Investigation; Nicholas M Luscombe, Resources, Supervision; Folkert J van Werven, Conceptualization, Resources, Formal analysis, Validation, Investigation, Writing – original draft, Writing – review and editing

## Author ORCIDs

Radhika A Varier ⓘ http://orcid.org/0000-0003-1302-3159
Theodora Sideri ⓘ http://orcid.org/0000-0002-5674-0804
Charlotte Capitanchik ⓘ http://orcid.org/0000-0001-9590-2792
Vincent WC Chan ⓘ http://orcid.org/0000-0002-6638-5498
Jernej Ule ⓘ http://orcid.org/0000-0002-2452-4277
Folkert J van Werven ⓘ http://orcid.org/0000-0002-6685-2084

## Decision letter and Author response

Decision letter https://doi.org/10.7554/eLife.84034.sa1
Author response https://doi.org/10.7554/eLife.84034.sa2

---

# Additional files

## Supplementary files

- Supplementary file 1. Yeast strains.
- Supplementary file 2. Plasmids.
- Supplementary file 3. Oligos sequences.
- Supplementary file 4. MS pulldown data.

- Supplementary file 5. iCLIP and miCLIP data table.
- Supplementary file 6. IP-MS data table.
- MDAR checklist

## Data availability

The miCLIP, iCLIP and RNA-seq RAW and processed data are available to review GEO accession GSE193561: https://www.ncbi.nlm.nih.gov/geo/query/acc.cgi?acc=GSE193561.

The following dataset was generated:

| Author(s) | Year | Dataset title | Dataset URL | Database and Identifier |
|---|---|---|---|---|
| Varier RA, van Werven F | 2022 | m6A reader Pho92 is recruited co-transcriptionally and couples translation efficacy to mRNA decay to promote meiotic fitness in yeast. | https://www.ncbi.nlm.nih.gov/geo/query/acc.cgi?acc=GSE193561 | NCBI Gene Expression Omnibus, GSE193561 |

The following previously published dataset was used:

| Author(s) | Year | Dataset title | Dataset URL | Database and Identifier |
|---|---|---|---|---|
| Brar GA, Yassour M, Friedman N, Regev A, Ingolia NT, Weissman JS | 2012 | High-resolution view of the yeast meiotic program revealed by ribosome profiling | https://www.ncbi.nlm.nih.gov/geo/query/acc.cgi?acc=GSE34082 | NCBI Gene Expression Omnibus, GSE34082 |

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
